# Efficient electrocatalytic acetylene semihydrogenation by electron-rich metal sites in N-heterocyclic carbene metal complexes

Lei Zhang [ID] [1], Zhe Chen [ID] [2], Zhenpeng Liu[1], Jun Bu[1], Wenxiu Ma[1], Chen Yan[1], Rui Bai[1], Jin Lin[1], Qiuyu Zhang[1], Junzhi Liu[3], Tao Wang [ID] [2✉] & Jian Zhang [ID] [1✉]

Electrocatalytic acetylene semihydrogenation is a promising alternative to thermocatalytic acetylene hydrogenation due to its environmental benignity and economic efficiency, but its performance is far below that of the thermocatalytic reaction because of strong competition from side reactions, including hydrogen evolution, overhydrogenation and carbon–carbon coupling reactions. We develop N-heterocyclic carbene–metal complexes, with electron–rich metal centers owing to the strongly σ–donating N-heterocyclic carbene ligands, as electrocatalysts for selective acetylene semihydrogenation. Experimental and theoretical investigations reveal that the copper sites in N-heterocyclic carbene–copper facilitate the absorption of electrophilic acetylene and the desorption of nucleophilic ethylene, ultimately suppressing the side reactions during electrocatalytic acetylene semihydrogenation, and exhibit superior semihydrogenation performance, with faradaic efficiencies of $\geq 98$ % under pure acetylene flow. Even in a crude ethylene feed containing 1 % acetylene ($1 \times 10^4$ ppm), N-heterocyclic carbene–copper affords a specific selectivity of >99 % during a 100-h stability test, continuous ethylene production with only ~30 ppm acetylene, a large space velocity of up to $9.6 \times 10^5$ mL·$g_{cat}^{-1}$·$h^{-1}$, and a turnover frequency of $2.1 \times 10^{-2}$ $s^{-1}$, dramatically outperforming currently reported thermocatalysts.

[1] Key Laboratory of Special Functional and Smart Polymer Materials of Ministry of Industry and Information Technology and Department of Advanced Chemical Engineering, School of Chemistry and Chemical Engineering, Northwestern Polytechnical University, 710129 Xi'an, China. [2] Center of Artificial Photosynthesis for Solar Fuels, School of Science, Westlake University, 310024 Hangzhou, China. [3] Department of Chemistry and State Key Laboratory of Synthetic Chemistry, The University of Hong Kong, Pokfulam Road, Hong Kong, China. ✉email: twang@westlake.edu.cn; zhangjian@nwpu.edu.cn

Ethylene, which is mainly derived from the steam cracking of liquefied petroleum gas, is the foremost olefin in the petrochemical industry, and its worldwide production capacity surpassed 170 million tons in 2016[1,2]. Unfortunately, ~1% acetylene is inevitably generated as an impurity in naphtha crackers used for ethylene production[3]. For the manufacture of diverse ethylene-based polymers, an acetylene impurity in the ethylene feedstock will seriously poison the polymerization catalysts (e.g., Ziegler–Natta catalysts) and thus adversely degrade the quality of the target polymers[4–6]. Currently, thermocatalytic acetylene semihydrogenation (TAH) is universally applied for hydrogenating acetylene impurities[7–11], but several major challenges remain: (i) the catalysts are rare and expensive Pd-based materials; (ii) relatively high temperature is imperative for improving the sluggish acetylene hydrogenation kinetics; (iii) excessive hydrogen gas ($H_2$) is involved as a hydrogen source; and (iv) side reactions such as overhydrogenation and carbon–carbon coupling occur concomitantly.

Thus, electrocatalytic acetylene semihydrogenation (EAH) under ambient conditions is a promising alternative strategy owing to its environmental benignity, operational simplicity, and economic efficiency[12–15]. For the EAH, water rather than $H_2$ gas serves as a hydrogen source. In the cathode, water molecules dissociate and offer active hydrogen for in situ hydrogenating acetylene ($C_2H_2 + 2H_2O + 2e^- \rightarrow C_2H_4 + 2OH^-$). Nevertheless, the development of EAH falls far behind conventional TAH as a result of poor solubility of acetylene in aqueous/organic solutions (1.06 g/kg $H_2O$) and strong competition of side reactions including hydrogen evolution reaction (HER), carbon–carbon coupling, and overhydrogenation. Despite using noble metals (e.g., Pd[16], Pt[17], and Ag[18]) as electrocatalysts in a pure acetylene atmosphere, the EAH still show inferior ethylene selectivity (<70%), very low current densities (<3.5 mA/cm²) and negligible space velocity (SV)[19,20]. Therefore, exploring nonnoble metal electrocatalysts with suppressed side reactions is of vital significance for EAH reactions.

In principle, acetylene molecules are electrophilic, while ethylene is nucleophilic[10]. Accordingly, to suppress the competitive HER, it is desired to enrich the electron density of active sites in the catalysts to increase the binding affinity of electrophilic acetylene[21,22]. Furthermore, benefiting from the nucleophilicity of ethylene, desorption of ethylene from the catalyst surfaces is preferred on electron-rich active sites in comparison with its overhydrogenation to ethane[23,24]. The carbon–carbon coupling of acetylene molecules can be avoided by spatially separating the active sites[10]. Thus, the design of electrocatalysts featuring individual electron-rich active sites is beneficial for selectively hydrogenating acetylene to ethylene.

Currently, owing to the strong σ-donation and robust chelation of N-heterocyclic carbene (1,3-bis(2,6-diisopropylphenyl)-1,3-dihydro-2H-imidazol-2-ylidene (NHC))[25–27], NHC–metal complexes have demonstrated outstanding potential for the catalysis of homogeneous organic reactions (e.g., alkyne hydrogenation[28–30], olefin metathesis[31]). The strong σ-donation of the NHC ligand considerably enriches the electron density of the chelated metal sites. Thus, to suppress side reactions during EAH reactions, NHC–metal complexes are promising candidate catalysts, but remain unexplored[32–34]. Here, we demonstrate NHC–metal complexes featuring electron-rich metal centers as electrocatalysts for selective acetylene hydrogenation (Supplementary Fig. 1; IUPAC names are given in the "Materials" subsection). Among a series of NHC–metal complexes (Cu, Ag, Au, and Pd), chloro[1,3-bis(2,6-diisopropylphenyl)imidazol-2-ylidene]copper(I) (NHC–Cu) presents ethylene faradaic efficiencies (FE_ethylene) of ≥98% under a pure acetylene stream and shows an ethylene-specific selectivity of >99% even in a crude ethylene feed

containing 1% acetylene during a 100-h stability test, suggesting its superb suppression of side reactions. Moreover, in a crude ethylene flow, NHC–Cu exhibits an SV of $9.6 \times 10^5$ mL/g_cat/h and a turnover frequency (TOF) of $2.1 \times 10^{-2}$ s$^{-1}$ and continuously outputs ethylene feedstock containing only ~30 p.p.m. acetylene, thus outperforming state-of-the-art thermocatalysts. In addition to these experimental results, in situ electrochemical Raman analyses and theoretical simulations reveal that the electron-rich Cu sites in NHC–Cu are beneficial for acetylene adsorption and ethylene desorption, ultimately improving the acetylene semihydrogenation kinetics.

## Results and discussion

**Screening of high-performance NHC–metal complexes.** Theoretical calculations were initially conducted to investigate the binding information, electron distributions and EAH kinetics of NHC–metal complexes (Pd, Au, Ag, and Cu). The negative binding energies between Pd (−2.15 eV), Au (−1.28 eV), Cu (−1.21 eV), and Ag (−0.55 eV) atoms and the NHC ligands clearly demonstrate strong chelation (Fig. 1a), indicating the excellent structural stability of the NHC–metal complexes. Afterwards, Bader charge analyses revealed that electrons of the NHC ligands were obviously donated to the metal sites: 0.15 e$^-$ for Pd, 0.12 e$^-$ for Ag, 0.27 e$^-$ for Au, and 0.10 e$^-$ for Cu (Fig. 1b and Supplementary Fig. 2), which unambiguously enriched the electron density of the metal sites. Because of the robust chelation and strong σ-donation of the NHC ligands, NHC–metal complexes are promising as potential EAH catalysts, but their electrocatalytic performance is also a result of the inherent electronic properties of their metal sites. Therefore, density functional theory (DFT) simulations were carried out to evaluate the EAH kinetics and competing HER on the NHC–metal complexes. As shown in Supplementary Table 1, the Gibbs free energies of hydrogen ($G_H$) are −1.23 eV for chloro[1,3-bis(2,6-diisopropylphenyl)imidazol-2-ylidene]gold(I) (NHC–Au), −1.13 eV for NHC–Cu, −0.82 eV for chloro[1,3-bis(2,6-diisopropylphenyl)imidazol-2-ylidene]silver(I) (NHC–Ag), and 0.51 eV for allyl[1,3-bis(2,6-diisopropylphenyl)imidazol-2-ylidene]chloropalladium(II) (NHC–Pd), which are far from zero. Especially, the large $G_H$ of NHC–Cu (−1.13 eV) means too strong H-adsorption and sluggish HER kinetics on Cu sites, which thus suppress the competitive HER reaction. Regarding the EAH kinetics, as indicated in Fig. 1c, acetylene adsorption, as expected, proceeds at the electron-rich metal sites of the NHC–metal complexes (Fig. 1c and Supplementary Figs. 4 and 5). Subsequently, the adsorbed acetylene molecules are hydrogenated to *CHCH₂ because of the low free energy barriers for NHC–Cu ($\Delta G = 0.02$ eV) and NHC–Pd ($\Delta G = 0.07$ eV) and the downhill energy changes for NHC–Ag ($\Delta G = -1.77$ eV) and NHC–Au ($\Delta G = -1.83$ eV). For the further hydrogenation of *CHCH₂ to form *CH₂CH₂, the free energy values of −1.39 eV for NHC–Cu and −1.40 eV for NHC–Pd indicate an exothermic process in comparison with the values of 0.56 eV for NHC–Au and 0.37 eV for NHC–Ag, which indicate an endothermic process. In principle, the selectivity of acetylene semihydrogenation or overhydrogenation is determined by ethylene desorption. As shown in Supplementary Fig. 4, CH₂CH₂* desorption is exothermic by 0.07 eV on the NHC–Cu complex. However, this step for NHC–Ag, NHC–Au, and NHC–Pd is endothermic by 0.20, 0.45, and 0.88 eV, respectively. In this respect, a more negative desorption free energy indicates a higher selectivity for ethylene, i.e., NHC–Cu (−0.07) > NHC–Ag (0.20) > NHC–Au (0.45) > NHC–Pd (0.88). Therefore, these theoretical results reveal that NHC–Cu is the most promising electrocatalyst for EAH.

Inspired by the appealing theoretical results obtained for the NHC–metal complexes, the EAH performance of the NHC–metal

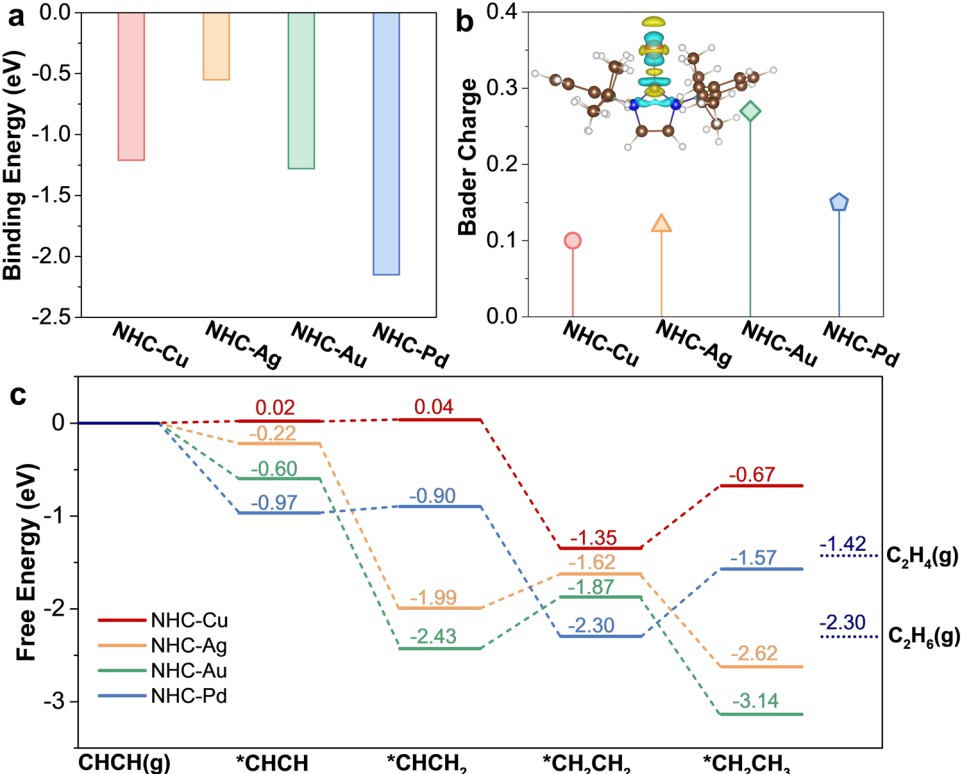

**Fig. 1 Theoretical calculations of NHC–metal complexes. a** Calculated binding energies ($E_b$, eV) between the NHC ligands and metal atoms in NHC–metal complexes. **b** Bader charge transfer from the NHC ligands to the metal sites. The inset displays the regions of charge accumulation (yellow) and depletion (cyan) in NHC–Cu. **c** Calculated free energy diagram of acetylene hydrogenation on NHC–metal complexes at a potential of 0 V vs. SHE.

complexes was experimentally evaluated in a 1 M KOH aqueous solution using a three-electrode flow cell fed with pure acetylene (1 cm²; Supplementary Figs. 6 and 7). Before the electrochemical tests, the chemical structures of the NHC–metal complexes were confirmed by utilizing proton nuclear magnetic resonance ([1]H NMR) spectra, X–ray diffraction (XRD) patterns, and X–ray photoelectron spectroscopy (XPS) spectra (Supplementary Figs. 8–11).

Linear sweep voltammetry (LSV) scans were recorded to assess the EAH activity of NHC–metal complexes (Fig. 2a). In contrast, NHC molecules and gas diffusion electrodes (GDEs) showed current densities of <5 mA/cm² at −0.7 V vs. RHE, which were far lower than the values of 65 mA/cm² for NHC–Cu, 55 mA/cm² for NHC–Ag, 44 mA/cm² for HHC–Pd, and 19 mA/cm² for NHC–Au. All of these potentials were relative to the RHE unless indicated otherwise. When NHC–Cu was subjected to a flow of argon gas, the LSV curve underwent a dramatically negative shift of >200 mV compared to that of the sample subjected to acetylene flow (Supplementary Fig. 12). These results clearly demonstrate the outstanding electrocatalytic activity of NHC–Cu toward acetylene.

Then, the faradaic efficiency (FE) distributions of the EAH products were investigated at different potentials. Notably, NHC–Cu retained a $FE_{ethylene}$ of ≥98%, with a negligible $FE_{C4}$ and no $FE_{ethane}$ over all applied potentials from −0.6 to −0.9 V (Fig. 2b and Supplementary Fig. 13). The total FE of the side reactions (HER, carbon–carbon coupling, and overhydrogenation) on NHC–Cu was <2% over all applied potentials, indicating an excellent suppression effect towards side reactions. Nevertheless, the $FE_{ethylene}$ values of NHC–Ag, NHC–Pd, and NHC–Au decreased dramatically with increasing potentials, suggesting the strong dependence of the $FE_{ethylene}$ on the applied potentials. Especially at a high potential of −0.9 V (Fig. 2c), the $FE_{ethylene}$

values decreased in the order of NHC–Cu (98%) > NHC–Ag (58%) > NHC–Au (52%) > NHC–Pd (26%). Furthermore, at −0.9 V, NHC–Cu achieved a partial ethylene current density of up to 159 mA/cm², which was substantially larger than the values of 64 mA/cm² for NHC–Ag, 37 mA/cm² for NHC–Pd, and 23 mA/cm² for NHC–Au (Fig. 2d).

To deeply probe the underlying mechanism of the superior EAH performance on NHC–Cu, electrochemical impedance spectroscopy (EIS) and in situ electrochemical Raman spectral analyses were conducted (Fig. 2e). NHC–Cu exhibited a lower charge transfer resistance (~45 Ω) than NHC–Ag (~69 Ω), NHC–Pd (~168 Ω), NHC–Au (>400 Ω), and NHC (>700 Ω), suggesting that it has a fast electron transfer process (Supplementary Fig. 14). As depicted in Fig. 2e, in comparison with those for bare NHC–Cu and NHC–Cu under the open-circuit potential, a Raman peak attributed to the $\nu(C\equiv C)$ stretching vibration appeared at 1954 cm⁻¹ for NHC–Cu at 0 V under acetylene flow[35,36], which confirmed acetylene adsorption on NHC–Cu[35,37]. Afterwards, the $\nu(C=C)$ stretching vibration of weakly π-bound ethylene at 1547 cm⁻¹ appeared gradually when the potential was increased from 0 to −0.8 V[36,38,39]. This Raman peak attributed to $\nu(C=C)$ disappeared after the EAH was terminated (Fig. 2e). These results confirmed the occurrence of EAH to form ethylene on NHC–Cu.

**EAH of NHC–metal complexes.** The EAH performance and structural stability of NHC–Cu in full cells are vital criteria for the practical implementation of these materials. To investigate these parameters, the cathodic EAH was coupled with the anodic oxygen evolution reaction (OER) on NiFe–LDH in a two-electrode flow cell (1 cm²). Under a pure acetylene flow, the current density of the full cell in a 1 M KOH aqueous solution reached 150 mA/cm² at a cell voltage ($E_{cell}$) of ~3.2 V (Fig. 3a). At current densities below

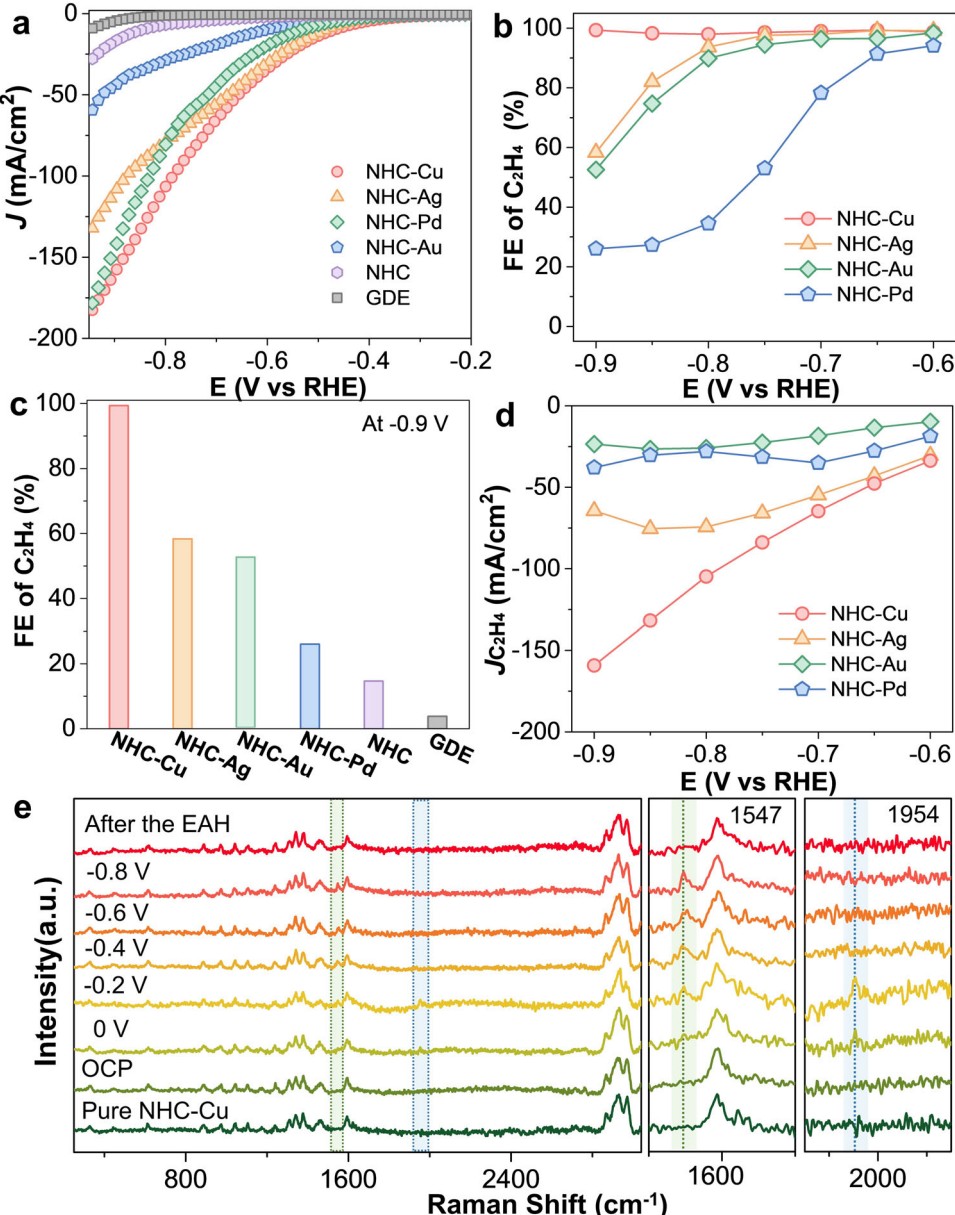

**Fig. 2 Experimental investigations of EAH on NHC–metal complexes. a** Polarization curves of NHC–metal complexes in a 1 M KOH aqueous solution under pure acetylene flow. **b** Ethylene FEs of NHC–metal complexes in a 1 M KOH aqueous solution at different potentials under pure acetylene flow. **c** Ethylene FEs for NHC–metal complexes, NHC, and gas diffusion electrodes at −0.9 V vs. reversible hydrogen electrode (RHE) in a 1 M KOH aqueous solution. **d** Partial current density to ethylene for NHC–metal complexes in a 1 M KOH aqueous solution at different potentials under pure acetylene flow. **e** In situ electrochemical Raman spectra of NHC–Cu in a 1 M KOH aqueous solution. For clarity, the spectral regions of 1500–1700 and 1850–2150 cm⁻¹ were expanded.

140 mA/cm², the FE$_{ethylene}$ of NHC–Cu was always ≥94 % (Fig. 3b). In particular, NHC–Cu displayed a FE$_{ethylene}$ of ~97% at 30 mA/cm² ($E_{cell}$ = −2.23 V). During the 80 h EAH stability measurement at 30 mA/cm² (Fig. 3c), the FE$_{ethylene}$ of NHC–Cu was consistent >96%. Notably, the FE of hydrogen on NHC–Cu was <2% over the 80−h EAH test, indicating outstanding HER suppression. Moreover, after the long−term EAH test, no structural variations of NHC–Cu were observed by means of XPS, ¹H NMR spectroscopy, and scanning electron microscopy (SEM) mapping (Supplementary Figs. 21–24). To further exclude the formation of 1,3-bis(2,6-diisopropylphenyl)imidazol-2-ylidene (hydroxy) copper (denoted as NHC–Cu(OH)) under alkaline conditions during the EAH test, ¹H NMR spectroscopy and EAH performance of presynthesized NHC–Cu(OH) were also

investigated (Supplementary Figs. 25–30). At −0.9 V vs. RHE, NHC–Cu(OH) exhibited a current density of 98 mA/cm² and a FE$_{ethylene}$ of 66%, which were substantially lower than 162 mA/cm² and 98% for NHC–Cu. These results reveal the strong chelation between the ligands and Cu sites.

Considering the low concentration of acetylene impurities in a practical ethylene stream from cracked gas, the influence of the gas partial pressure on the EAH performance of NHC–Cu was then investigated in a 1 M KOH aqueous solution fed a mixture of argon and acetylene. The LSV curves of NHC–Cu display a distinctly positive shift with an increase in the acetylene concentration (Supplementary Fig. 31). The FE$_{ethylene}$ at −0.6 V decreased from 98% for pure acetylene to 70% for 1% acetylene (Supplementary Fig. 32) due to the enhanced surface hydrogen coverage.

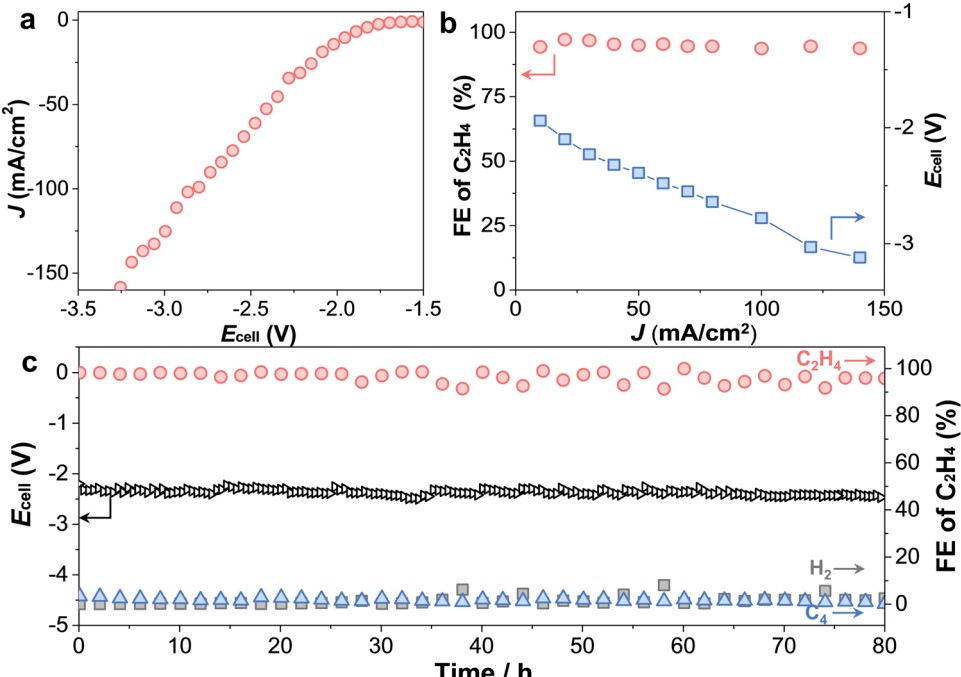

**Fig. 3 The EAH performance of NHC–Cu in a two-electrode flow cell. a** LSV curve of NHC–Cu in a 1 M KOH aqueous solution fed with pure acetylene.
**b** Ethylene FEs and corresponding $E_{cell}$ as a function of current densities. **c** Long−term EAH durability test of NHC–Cu at 30 mA/cm².

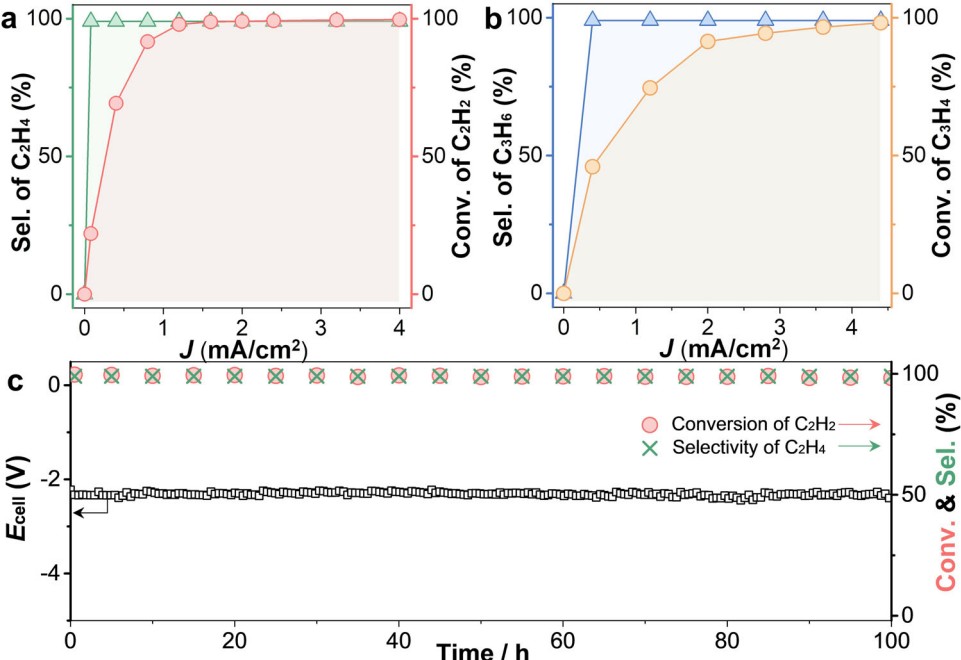

**Fig. 4 EAH performance of NHC–Cu in crude olefins using a large two-electrode flow cell (25 cm²). a** Acetylene conversion and specific selectivity of ethylene as a function of current density at a flow rate of 10 mL/min in crude ethylene containing 1% acetylene. **b** Propyne conversion and specific selectivity of propylene as a function of current density at a flow rate of 10 mL/min in propylene containing 1% propyne. **c** The 100–h EAH durability test of NHC–Cu at 4 mA/cm².

To satisfy the requirements for the industrial production of polymer-grade ethylene, the EAH performance of 1% acetylene impurities ($1 \times 10^4$ p.p.m.) in crude ethylene was assessed in a 1 M KOH aqueous solution at a gaseous flow rate of 10 mL/min by using a large two-electrode flow cell (25 cm², Supplementary Fig. 33). Figure 4a displays the acetylene conversion and ethylene selectivity of NHC–Cu as a function of the current density. Here,

the specific selectivity of ethylene refers to the percentage of acetylene that is hydrogenated into ethylene (Supplementary Fig. 35). Below 2 mA/cm², the acetylene conversion was >99%, and the related specific selectivity of ethylene was maintained at >99%. When the current density was ≥2 mA/cm², NHC–Cu achieved an acetylene conversion of 99.7% and an ethylene–specific selectivity of >99%, thus outputting high-

**Table 1 Comparison of acetylene semihydrogenation performance between EAH of NHC–Cu and the reported TAH of thermocatalysts.**

| Catalyst | Feed gas composition | T (°C) | SV (mL/g$_{cat}$/h) | TOF (s$^{-1}$) | Specific selectivity (%) | Residual C$_2$H$_2$ (p.p.m.) | Stability (h) | Refs. |
|---|---|---|---|---|---|---|---|---|
| NHC–Cu | C$_2$H$_2$/C$_2$H$_4$ (1/99) | 25 | $9.6 \times 10^5$ | $2.1 \times 10^{-2}$ | >99 | ~30 | 100 | This work |
| Cu$_1$/ND@G | C$_2$H$_2$/C$_2$H$_4$/H$_2$/He (1/20/10/69) | 200 | $3 \times 10^3$ | $1.7 \times 10^{-3}$ | >98 | 500 | 60 | 48 |
| Ni$_3$Ga | C$_2$H$_2$/C$_2$H$_4$/H$_2$/He (0.5/50/10/39.5) | 200 | $4 \times 10^4$ | $1.4 \times 10^{-6}$ | 77 | 500 | 50 | 49 |
| Al$_{13}$Fe | C$_2$H$_2$/C$_2$H$_4$/H$_2$/He (0.5/50/5/44.5) | 200 | $9 \times 10^4$ | / | 82 | 1000 | 20 | 50 |
| Pd/PPS | C$_2$H$_2$/C$_2$H$_4$/H$_2$/C$_3$H$_8$/N$_2$ (0.6/49.3/0.9/0.6/48.6) | 100 | $3.26 \times 10^2$ | / | >65 | 60 | 200 | 51 |
| Pd@SOD | C$_2$H$_2$/H$_2$/N$_2$ (0.6/6/93.4) | 150 | $3 \times 10^4$ | / | 94.5 | 12 | 50 | 7 |
| Single-atom Pd | C$_2$H$_2$/C$_2$H$_4$/H$_2$/He (0.5/50/5/44.5) | 120 | $1.2 \times 10^3$ | / | 93.4 | 200 | 3 | 52 |
| PdZn@ZIF-8C | C$_2$H$_2$/C$_2$H$_4$/H$_2$/Ar (0.65/50/5/44.35) | 115 | $4.8 \times 10^4$ | / | 80 | 2200 | 12 | 53 |
| Pd-Pt/SiO$_2$ | C$_2$H$_2$/C$_2$H$_4$/H$_2$ (1.67/3.33/95) | 80 | $1.35 \times 10^5$ | / | 40 | 167 | / | 8 |
| Ni$_3$ZnC$_{0.7}$ | C$_2$H$_2$/C$_2$H$_4$/H$_2$/He (0.5/20/4.5/75) | 200 | / | / | 93 | 50 | 12 | 54 |
| Graphene | C$_2$H$_2$/C$_2$H$_4$/H$_2$/N$_2$ (0.1/0.9/3/94) | 110 | $1.46 \times 10^4$ | / | 92 | 210 | / | 55 |

quality ethylene feedstock with only ~30 p.p.m. acetylene impurities. For crude ethylene, NHC–Cu still exhibited an extremely high TOF value of $2.1 \times 10^{-2}$ s$^{-1}$ at 4 mA/cm$^2$, which was increased by more than one order of magnitude in comparison with those of previously reported thermocatalysts (e.g., $1.4 \times 10^{-6}$ s$^{-1}$ for Ni$_3$Ga and $1.7 \times 10^{-3}$ s$^{-1}$ for Cu$_1$/ND@G, etc.) (Table 1). Next, during a 100–h stability test in a crude ethylene flow, both the acetylene conversion and specific selectivity of ethylene of NHC–Cu remained >99% at 4 mA/cm$^2$ (Fig. 4c and Supplementary Fig. 36). Accordingly, NHC–Cu afforded a very large SV of up to $9.6 \times 10^5$ mL/g$_{cat}$/h, which was one order of magnitude higher than those of current thermocatalysts (e.g., $3 \times 10^3$ mL/g$_{cat}$/h for Cu$_1$/ND@G and $9 \times 10^4$ mL g$_{cat}$/h for Al$_{13}$Fe, etc.) (Table 1). Noticeably, over long-term EAH operation, no ethane and negligible C$_4$ products were detected (Supplementary Fig. 36), proving that overhydrogenation and carbon–carbon coupling reactions were intrinsically suppressed even at low concentrations of acetylene. In addition, when electrocatalytic hydrogenation was applied for propyne impurities in propylene, NHC–Cu exhibited a conversion of ~98% and a propylene-specific selectivity of >99% at 4.4 mA/cm$^2$ (Fig. 4b and Supplementary Fig. 37). Therefore, NHC–Cu shows promise for application in selective catalysis of alkyne semihydrogenation.

In summary, we explored NHC–metal complexes featuring electron-rich metal sites as alkyne semihydrogenation electrocatalysts by tailoring the adsorption/desorption of the reactants and products. The electron–rich copper sites in the NHC–Cu complex promoted the adsorption of electrophilic acetylene and the desorption of nucleophilic ethylene, which inherently suppressed the side reactions during EAH. As a result, NHC–Cu achieved state–of–the–art ethylene FE, specific selectivity, electrocatalytic stability, SV, and TOF. Therefore, this work not only opens a fresh window for NHC–metal complexes in the field of electrocatalysis but also provides insight into the design of selective hydrogenation catalysts.

## Methods
**Materials**. NHC, NHC–Cu, NHC–Au, NHC–Ag, and NHC–Pd were all purchased from Sigma–Aldrich. Acetone (≥99.5%), deuterated acetone (99.5%), ethanol

(99%), nickel nitrate hexahydrate (Ni(NO$_3$)$_2$·6H$_2$O, 98%), urea and iron(III) nitrate (Fe(NO$_3$)$_3$·9H$_2$O, 99%) were obtained from Aladdin (China). Ni foam was purchased from KunShan Kuangxun Ltd. (China). The carbon paper with a gas diffusion layer (Sigracet 29 BC) was purchased from the Fuel Cell Store. Nafion solution (5 wt%) was purchased from Shanghai Hesen (China). Ultrapure water was distilled via a High-Q distillation system (>18.25 MΩ).

**Synthesis of NiFe-LDH on nickel foam**. Nickel (Ni) foam was first washed with 3 M HCl, ethanol, and deionized water. Then, this pretreated Ni foam with a size of $1.5 \times 3$ cm$^2$ was immersed into a Teflon autoclave with a 36 mL aqueous solution containing 0.5 mmol of Ni(NO$_3$)$_2$·6H$_2$O, 0.5 mmol of Fe(NO$_3$)$_3$·9H$_2$O, and 2 mmol of urea. Afterwards, the sealed autoclave was transferred into an oven at 120 °C for 12 h. Finally, the NiFe–LDH anchored on the Ni foam was collected with a loading weight of ~1 mg/cm$^2$ after cooling to room temperature and washing with deionized water.

**Synthesis of NHC–Cu(OH)**. The compound NHC–Cu(OH) was synthesized according to the previous reference[40]. Specifically, 100 mg NHC–Cu (0.205 mmol) was added into a dried round bottle (25 mL) and transferred to the glovebox. After adding anhydrous cesium hydroxide (61 mg, 0.410 mmol) and dry tetrahydrofuran (4 mL) into the above bottle, the mixture was reacted at room temperature for 8 h under N$_2$ atmosphere. The resulting solution was filtered through a plug of Celite and concentrated in vacuo until a white precipitate was formed (ca. 1 mL remaining). Fifty micrograms (52%) of NHC–Cu(OH) was finally collected after the precipitation in hexane and dried under vacuum.

**$^1$H NMR of [NHC–Cu(OH)] (500 MHz, CD$_2$Cl$_2$)**. δ 7.52 (t, $J$ = 8.0 Hz, 2H), 7.33 (d, $J$ = 8.0 Hz, 4H), 7.13 (s, 2H), 2.57 (p, $J$ = 7.0 Hz, 4H), 1.27 (d, $J$ = 7.0 Hz, 12H), and 1.21 (d, $J$ = 7.0 Hz, 12H).

**$^1$H NMR of [NHC–Cu] (500 MHz, CD$_2$Cl$_2$)**. δ 7.54 (t, $J$ = 8.0 Hz, 2H), 7.35 (d, $J$ = 8.0 Hz, 4H), 7.19 (s, 2H), 2.57 (p, $J$ = 7.0 Hz, 4H), 1.29 (d, $J$ = 7.0 Hz, 12H), and 1.23 (d, $J$ = 7.0 Hz, 12H).

**Preparation of the cathode**. Catalyst ink at a concentration of 0.25 mg/mL was prepared by dispersing 10 mg of catalyst in a mixture of 39.985 mL of acetone and 15 μL of Nafion (5 wt%). For a 1 cm$^2$ electrode area, 100 μL of ink containing 25 μg of catalyst was sprayed onto a GDE and dried thoroughly. For a 25 cm$^2$ electrode area, 2.5 mL of ink containing 250 μg of catalyst was sprayed onto a GDE and dried thoroughly. The final loading of the catalyst was ~25 μg/cm$^2$ after normalization to the electrode area of the cathode.

**Design of the flow cells**. Measurements of the EAH were conducted in a customized flow cell containing housings, flow plates, an anodic chamber, an anion–exchange membrane, a cathodic chamber and a gaseous chamber.

Specifically, one anodic chamber was used for water oxidation, one cathodic chamber was used for acetylene hydrogenation, and one gaseous chamber was used for gas diffusion. Except for the gaseous chamber, a 1 M KOH aqueous solution as the electrolyte was refluxed in each chamber driven by a peristaltic pump at a rate of 10 mL/min. An anion–exchange membrane was employed to separate the anodic and cathodic chambers. The above components were tightened to form a sealed flow cell by applying silicone gaskets and bolts. Two flow cells with different electrode areas were fabricated for the EAH: 1 cm$^2$ (0.5 × 2 cm$^2$) and 25 cm$^2$ (5 × 5 cm$^2$).

**Electrochemical measurements**. All electrochemical measurements were conducted using a CH Instrument 760E Potentiostat. Three–electrode flow cells were equipped with saturated Hg/HgO as the reference electrode, Ni foam as the counter electrode, and a GDE coated with catalysts as the working electrode. All potentials presented in this system were referenced to RHE using the equation of $E_{RHE} = E_{Hg/HgO} + 0.099\ V + 0.059\ V \times pH$. Two–electrode flow cells were equipped with a NiFe–LDH on Ni foam as the anodic electrode and a GDE coated with catalysts as the cathodic electrode. The potentials presented in this system were cell voltages ($E_{cell}$).

All the EAH experiments were performed at room temperature. The flow rate of acetylene was 20 mL/min for all experiments unless mentioned otherwise. Cyclic voltammetry of the catalysts was carried out in a 1 M KOH aqueous solution at a scan rate of 100 mV/s under acetylene flow. LSV scans were performed in a 1 M KOH aqueous solution fed with acetylene at a scan rate of 1 mV/s. The current density was normalized with the geometric area of the electrode. The amperometric curves ($i–t$) were executed at different potentials for a fixed time to evaluate the reaction products. Chronopotentiometry was carried out at a certain current to assess the stability of the catalysts. The gaseous products of EAH were analyzed using gas chromatography (GC). EIS of the catalysts was conducted in a 1 M KOH aqueous solution purged with acetylene at −0.4 V over a frequency range of 10 kHz to 0.01 Hz.

**FEs of the products**. The FEs of the gaseous products were calculated according to the following equation:

$$FE = \frac{A_g \times R_g \times e_g \times F}{I \times V} \tag{1}$$

Here, $A_g$ is the amount (p.p.m.) of gaseous products integrated from the peak area using the standard calibration curves. $R_g$ is the flow rate of the gas (mL/min). $e_g$ is the number of electrons transferred to the gas product. $F$ is Faraday's constant (96,485 C/mol). $I$ is the total current for EAH. $V$ is the molar volume of a gas (22.4 L/mol).

**Specific selectivity of ethylene**. The specific selectivity of ethylene was estimated based on the number of carbon atoms according to the following equation:

$$\text{Specific selectivity of ethylene} = \frac{C_{ethylene}}{^0C_{acetylene}} \times 100\% = \frac{^0C_{acetylene} - C4 - C_{ethane}}{^0C_{acetylene}} \times 100\% \tag{2}$$

Here, $C_{ethylene}$ represents the number of carbon atoms in the ethylene product; $^0C_{acetylene}$ is the number of carbon atoms in acetylene in the crude ethylene stream; C4 and $C_{ethane}$ are the numbers of carbon atoms in the C4 and ethane products, respectively. To calculate the specific selectivity of ethylene, it was assumed that acetylene was hydrogenated to ethylene, ethane and the C4 product. No hydrocarbons higher than C4 were observed in the GC analyses.

**Calculation of the TOF and SV**. The TOF (s$^{-1}$) of the catalysts was calculated according to the following equation:

$$TOF = \frac{n_{ethylene}}{n_{catalyst} \times t} \tag{3}$$

Here, $n_{ethylene}$ (mol) is the mole number of ethylene in the kinetic region, as quantified by GC analyses. $n_{catalyst}$ is the mole number of catalysts involved in EAH. $t$ (s) is the time of EAH for the generation of ethylene.

The SV (mL·g$_{cat}^{-1}$·h$^{-1}$) of the catalysts was calculated based on the amount of feed gas according to the following equation:

$$SV = \frac{V_{gas}}{m_{catalyst} \times t} \tag{4}$$

Here, $V_{gas}$ (mL) is the volume of feed gas involved in EAH, as monitored by a gas flowmeter. $m_{catalyst}$ (g) is the mass of the catalysts sprayed on the GDE. $t$ (h) is the time of EAH under a flow of crude ethylene containing 1% acetylene impurities.

**Analyses of the catalyst after long–term EAH**. To analyze the catalysts after EAH, the cathode after long–term EAH was investigated directly by XPS and SEM characterizations. The NMR spectra of the catalysts after long–term EAH were obtained by peeling the catalysts from the cathode into deuterated acetone.

**DFT calculations**. The Vienna Ab Initio Simulation Package code was used to perform all spin–polarized DFT calculations[41]. The revised Perdew–Burke–Ernzerhof functional was employed to describe the exchange-correlation interactions within the generalized gradient approximation[42,43]. The electron–ion interactions were represented by the projector augmented wave method[44]. The kinetic energy cutoff of the plane wave was set to 400 eV, and the convergence criteria for the residual forces and total energies were set to 0.02 eV/Å and 10$^{-5}$ eV, respectively. The empirical correction in Grimme's method (DFT + D3) was adopted to describe the van der Waals interactions[45]. Transition state pathways were determined with the climbing image-nudged elastic band method[46].

The structures of all the NHC–metal catalysts were determined with a vacuum space of 20 Å in three directions to minimize the interaction between periodic images. A 1 × 1 × 1 Monkhorst–Pack $k$-point mesh was used to sample the Brillouin zone. To examine the structural stability of the NHC–metal complexes, we computed the binding energies ($E_b$, eV) of all the systems according to the equation $E_b = E_{NHC-metal} - E_{NHC} - E_{metal}$, where $E_{NHC-metal}$ and $E_{NHC}$ are the total energies of pristine NHC with and without metal atoms, respectively, and $E_{metal}$ is the total energy of isolated single metal atom.

The change in the Gibbs free energy of the reaction ($\Delta G$) for each elementary step during the acetylene hydrogenation process was calculated by using the computational hydrogen electrode model proposed by Nørskov et al.[47]. Given that the Gibbs free energies of hydrogen on the NHC–Cu catalyst is far from zero, the preadsorption of hydrogen onto the catalysts was considered during the acetylene hydrogenation simulation. The chemical potential of the proton–electron pair in an aqueous solution is related to that of one-half of the chemical potential of an isolated hydrogen molecule. Based on this model, the $\Delta G$ value can be obtained by the formula $\Delta G = \Delta E + \Delta ZPE - T\Delta S + \Delta G_{pH} + eU$, where $\Delta E$ is the reaction energy of the reactant and product species adsorbed on the catalyst, which was obtained directly from DFT computations; $\Delta ZPE$ and $\Delta S$ are the changes in the zero point energy and entropy between the adsorbed species and the gas–phase molecules at 298.15 K, which were calculated from the vibrational frequencies. $\Delta G_{pH}$ is the free energy correction of pH and was calculated by $\Delta G_{pH} = K_B T \times pH \times \ln 10$. Notably, the pH value was set to zero in this work for simplicity; $U$ was the applied potential.

**Characterizations**. Transmission electron microscopy was carried out on an FEI Talos F200X at an acceleration voltage of 200 kV. SEM and energy-dispersive X-ray mapping were performed on a field–emission FEI–Verios G4 microscope operating at 15 kV. $^1$H NMR spectra were obtained on a Bruker Ascend 400 MHz spectrometer by using deuterated acetone as the reagent. XRD patterns were recorded on a Rigaku SmartLab Powder X–ray diffractometer equipped with a Cu Kα X–ray source and a D/teX Ultra detector, with a scanning rate of 10°/min over a range of 5–50° (2$\theta$). XPS was performed using a Kratos AXIS Ultra DLD equipped with Al Kα radiation. The binding energy of the C 1s peak (284.6 eV) was employed as a standard to calibrate the binding energies of other elements. In situ Raman spectroscopy was carried out on a Renishaw inVia confocal microscope using an objective with a focal distance of 15 mm, 1800 l/mm grafting, and 2 mW laser intensity with 514 nm laser excitation. All Raman spectra were collected until the current remained unchanged.

## Data availability

The data that support the findings of this study are available from the corresponding author upon reasonable request. Source data are provided with this paper.

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

## Acknowledgements

This work was supported by the Fundamental Research Funds for the Central Universities (Grant No. 310201911cx028), the Natural Science Foundation of Shaanxi Province (No. 2020JQ–141), and the National Natural Science Foundation of China (No. 22005245). L.Z. acknowledges support from the startup fund (No. D5000210090) of Northwestern Polytechnical University, the National Natural Science Foundation of China (No. 52101271). the Natural Science Foundation of Shaanxi Province (No. 2021JQ–094), the fellowship of China Postdoctoral Science Foundation (2021M692619), and the Regional Joint Fund of Guangdong Province (No. 2020A1515111017). J.L. is grateful for the funding from the Hong Kong Research Grants Council (HKU 27301720) and ITC to the SKL. We thank Lin Cheng and Prof. Hanchen Liu at the School of Science, Xi'an Polytechnic University, for help with the Raman experiments. We also thank Chang Wang for the synthesis of NHC–Cu(OH) at the University of Hong Kong. T.W. acknowledge the start-up fund of Westlake University and Westlake University HPC Center for computation support.

## Author contributions

L.Z. and J.Z. conceived and designed the experiments. L.Z. carried out the experiments and analyzed the results. Z.L. performed the characterizations of the catalysts. J.B. optimized the electrochemical and product analysis setups. W.M., C.Y., R.B., and Ji.L. contributed to the electrochemical measurements. Z.C. and T.W. conducted the DFT

calculations and data analysis. J.L and Ju.L. conducted the synthesis of NHC–Cu(OH). L.Z. and J.Z. cowrote the manuscript. Q.Z. and other authors discussed the results and commented on the manuscript.

## Competing interests

The authors declare no competing interests.
