## [Peer Review File · Nature Communications]

Title: Efficient electrocatalytic acetylene semihydrogenation on electron-rich metal sites in N-heterocyclic carbene metal complexesREVIEWER COMMENTS

Reviewer #1 (Remarks to the Author):

Reviewer Report

TITLE: Efficient electrocatalytic acetylene semihydrogenation on electron-rich metal sites in N-heterocyclic carbene metal complexes

AUTHORS: Lei and co-workers

SUBMITTED TO: Nature Communications

This paper describes the electrocatalytic partial hydrogenation of acetylene over N-heterocyclic carbene metal complexes under alkaline conditions. The development and understanding of PGM-free catalyst for this reaction are important goals, and this work represents an effort to this challenge. The authors conducted calculations, which were further validated with experiments showing that NHC-Cu is highly active and selective for electrochemical partial hydrogenation to ethylene. This manuscript is interesting and relevant for the readership of Nature Communications. If the authors can address the following problems, I believe the manuscript should be publishable:

1. In industry, when ethylene is produced by steam cracking of naphtha, the resultant gas feed has high temperature of ca. 350-400°C, and therefore, has to be chilled down to ca. 200°C to conduct thermocatalytic partial hydrogenation of C₂H₂ to C₂H₄ over Pd/Ag catalyst. The authors conducted electrochemical reaction at RT, which make overall process more demanding in terms of energy, since the gas feed from cracking has to be chilled down to even lower temperature than in the case of thermocatalytic hydrogenation, namely to RT. In this particular case, I would say that hydrogenation at elevated temperatures would be more advantageous in terms of saving on chilling energy to conduct electrocatalytic acetylene partial hydrogenation. Please provide some rationale.
2. N-heterocyclic carbene metal complexes are well-known catalysts for olefin metathesis, did the authors observe any signature of this competitive reaction?
3. I am not well-familiar with the calculations, but the experiments show a clear difference between NHC-Cu and other metal complexes in terms of partial hydrogenation of acetylene. The calculations suggest, however, only the activity of NHC-Cu and selectivity for ethylene. What is needed, in my view, and what I think is buried in the calculations that have already been done, is a theoretical determination of the relative selectivity of NHC-Ag, NHC-Au, NHC-Pd as well as NHC-Cu. Specifically, I would like to see the initial state and the zero of energy to be the un-adsorbed gas phase reactant and the final state to be the desorbed product. This is important to evaluate the relative energetics between the electrocatalysts which will have different reactant adsorption energies.
4. What is the reason to select alkaline electrolyte and is electrophilic/nucleophilic character of the substrates/intermediates/products affected by the electrolyte (e.g., deprotonation at pH=14). The same question to the effect of electrolyte on electron-rich/poor character of the active sites as well as on σ -donation of the NHC ligand.
5. What is the exact reason for a lower charge transfer resistance of NHC-Cu among the other cathodes?

6. Which materials has been used as GDE?
7. Did the authors study the leaching of the catalyst from GDE to the electrolyte solution while conducting stability testing?
8. The main problem of the paper is that this paper is lack of discussion. As the authors mentioned in introduction, there are many different catalysts for gas-phase partial hydrogenation of acetylene with interesting performance known. Unfortunately, there is no information about electrocatalytic hydrogenation of acetylene. It is not clear if such electrochemical efforts are not exist or have been largely ignored by authors. The authors should show what are the new insights beyond the published results.
9. The suppression of the competitive HER reaction should be discussed in more details.
10. The origin of hydrogen for electrochemical hydrogenation of acetylene should be discussed in more details as well.

Review Sent Date: 13-May-2021

Reviewer #2 (Remarks to the Author):

I have read the paper very much from the angle of trying to understand the underpinning chemistry rather than comment on the catalytic activity of the reported system relative to prior examples - other referees are better equipped to do that than me. I am left very confused by what the authors think is going on in their system by their continual reference to NHC-Cu, NHC-Ag, NHC-Au and NHC-Pd. In all cases, they start with NHC-M-chloride complexes; to my mind placing these in aqueous KOH would undoubtedly afford metal hydroxide complexes. Indeed, for an overview of this, see: D.J. Nelson, S.P. Nolan, *Coord. Chem. Rev.* 2017), 353, 278–294.

This review and the primary literature shows that exactly these types of species have been produced for the specific compounds being studied. For Cu, see: G. C. Fortman, A. M. Z. Slawin, S. P. Nolan, *Organometallics* 2010, 29, 38966-3972. For Au, see: S. Gaillard, A. M. Z. Slawin, S. P. Nolan, *Chem. Commun.*, 2010, 46, 2742–2744. For a very close Pd analogue, see: J. D. Egbert, A. Chartoire, A. M. Z. Slawin, S. P. Nolan, *Organometallics* 2011, 30, 4494–4496.

Why is this important? Simply because copper is what the authors focus their attention on due to its high reactivity. A study of the (IPr)CuOH complex for efficient homogeneous catalytic alkyne semihydrogenation has been reported previously: N. O. Thiel, J. F. Teichert, *Org. Biomol. Chem.*, 2016, 14, 10660–10666.

I cannot find any of the references above to show that the authors appreciate the fundamental chemistry that could help to explain their findings. They should repeat their studies using isolated examples of the (NHC)M(OH) precursors to show whether these display the same activity. Moreover, as apparent from the Nelson/Nolan review above, studies of stoichiometric alkyne deprotonation by e.g. (IPr)AuOH have been found to afford the corresponding (NHC)M(alkynyl) complexes. These too should be employed as catalytic precursors and their activity compared.

In short, the work should be rejected until the authors have taken on board the basic reactivity known in the literature to be associated with their catalyst precursors, appropriately cited the literature (the Org. Biomol. Chem. paper is particularly pertinent) and carried out tests of the activity of these (NHC)M(OH)/(NHC)M(alkynyl) species. Until that is done, it is my view that the work is not suitable for publication anywhere as I believe it to be fundamentally flawed in its current format.

Reviewer #3 (Remarks to the Author):

In this manuscript, the author introduced an N-heterocyclic carbene-Cu complex (NHC-Cu) as an efficient electrocatalyst for acetylene semihydrogenation reaction. Benefitting from the strong σ -donation from NHC, their catalyst exhibited outstanding activity, selectivity and durability for acetylene semihydrogenation reaction. This is an interesting attempt by employing electrons and protons to selectively convert C₂H₂ to C₂H₄. The authors have performed comprehensive experiments to test this concept. However, major flaws exist. I would suggest its publication after a major revision to address the following comments.

1. minor wording: you mentioned "High temperature" as a challenge for acetylene thermocatalytic semihydrogenation reaction (TAH). However, in literature, the reaction temperature for TAH is usually 80 – 100 °C. I don't think it's "high temperature". This needs to be reworded. Comparing with electrocatalysis at ambient Ts, TAH does need some thermal energy input; but definitely not "high T".

2. Comments regarding the free energy diagram: it seems to me that according to the Bader Charge analysis, Pd-NHC should be more electrophilic, thus should adsorb C₂H₂ more easily. Why in the free energy diagram, the first step of C₂H₂ adsorption, it's not the most down-hill on Pd-NHC?

What is the RDS on M-NHC? The first adsorption step is not favored on Cu-NHC while the last desorption step of C₂H₄ seems the most facile on Cu-NHC. This calculation result combined with the experimental observation of the best performance of Cu-NHC may suggest the desorption might be the RDS. However, it needs some evidence and clarification. It seems kinetic experiments have been performed but not be analyzed in-depth to provide insight into the RDS. I would suggest analyzing the kinetics more carefully and obtain reaction rates/RDS which can be combined with DFT calculations.

The reaction profile seems to be solely based on thermodynamic calculations without considering electrode polarization. I would suggest adding a free energy diagram with potential consideration.

3. Comments on the electrocatalysis experimentation:

The Raman spectroscopy of Cu-NHC can be compared with other M-NHC just to confirm whether the assumed adsorption/desorption and the difference among M-NHCs echo with the DFT calculations. A side note: Why does the peak at 1954 (indication of C₂H₂ adsorption) disappear after -0.4 V?

How are those current densities normalized? Is it based on geometric surface area or ECSA?

For the “1 % acetylene impurities (1×10^4 ppm) in crude ethylene” test, I noticed the flow rate of feeding gas is 10 sccm. However, in literature, THA catalysts typically work at a flow rate of 50-100 sccm. Have you tried a higher flow rate of feed gas here? The selectivity probably won't change much but I am wondering whether the HNC-Cu system can still keep that high conversion.

What about the solubility issue? Will the solubility of C_2H_2 in water become a concern for the electronic semi-hydrogenation of C_2H_4 ? This is an issue where THA catalysts do not suffer from.

Response to Reviewer 1:**Comments:**

This paper describes the electrocatalytic partial hydrogenation of acetylene over N-heterocyclic carbene metal complexes under alkaline conditions. The development and understanding of PGM-free catalyst for this reaction are important goals, and this work represents an effort to this challenge. The authors conducted calculations, which were further validated with experiments showing that NHC-Cu is highly active and selective for electrochemical partial hydrogenation to ethylene. This manuscript is interesting and relevant for the readership of Nature Communications. If the authors can address the following problems, I believe the manuscript should be publishable:

Response:

We greatly thank the Reviewer for above positive comments on our work.

Question 1:

In industry, when ethylene is produced by steam cracking of naphtha, the resultant gas feed has high temperature of ca. 350–400 °C, and therefore, has to be chilled down to ca. 200 °C to conduct thermocatalytic partial hydrogenation of C₂H₂ to C₂H₄ over Pd/Ag catalyst. The authors conducted electrochemical reaction at RT, which make overall process more demanding in terms of energy, since the gas feed from cracking has to be chilled down to even lower temperature that in the case of thermocatalytic hydrogenation, namely to RT. In this particular case, I would say that hydrogenation at elevated temperatures would be more advantageous in terms of saving on chilling energy to conduct electrocatalytic acetylene partial hydrogenation. Please provide some rationally.

Response:

We appreciate the Reviewer's comment. As the Springer book of "Catalytic and Process Study of the Selective Hydrogenation of Acetylene and 1,3-Butadiene" mentioned, industrial processes for acetylene semihydrogenation are typically front-end and tail-end processes. For the front-end process, since hydrogenation reactor locates before the demethanizer, gas stream from the cracker directly flows into

the hydrogenation reactor for the semihydrogenation after chilling down. In this case, as demonstrated by the Reviewer, hydrogenation process at elevated temperatures would be more advantageous in terms of saving on chilling energy. However, the feed gas of hydrogenation reactor in the front-end process contains not only C₂ species, but also H₂, CH₄, CO, etc. The nonadjustable H₂/C₂H₂ ratio and the existence of other gases unavoidably result in low acetylene conversion and ethylene selectivity even using noble-metal Pd-based catalysts. Accordingly, the tail-end process is further developed.

For tail-end process, the hydrogenation reactor locates after the demethanizer and deethanizer. The C₂ species are first separated out from crude gas stream from the cracker through the demethanizer and deethanizer. Then, a certain amount of H₂ is mixed with C₂ stream into the hydrogenation reactor. Noticeably, the operating temperatures of demethanizer and deethanizer are below room temperature. So, in the tail-end process, the cooled C₂ stream has to be heated up to ca. 200 °C for thermocatalytic acetylene hydrogenation, which causes excessive energy consumption. Therefore, for tail-end process, electrocatalytic acetylene semihydrogenation at room temperature is more advantageous in terms of energy consumption than thermocatalytic acetylene semihydrogenation.

Question 2:

N-heterocyclic carbene metal complexes are well-known catalysts for olefin metathesis, did the authors observed any signature of this competitive reaction?

Response:

As demonstrated by the Reviewer, N-heterocyclic carbene metal complexes are well-known catalysts for olefin metathesis. As a traditional example of olefin metathesis (Fig. R1), symmetrical disubstituted olefins (each with blue or red substituents) are converted into the unsymmetrical olefins (with one blue and one red substituents). Here, if olefin metathesis occurs during the electrocatalytic acetylene semihydrogenation, ethylene is only converted into ethylene due to the H substituent of ethylene. Therefore, it is difficult to observe the olefin metathesis in this work. In

addition, the olefin metathesis generally proceeds in organic solvents (e.g., DCM, THF, C₆H₆, etc.) (*Chem. Rev.*, 2009, 109, 3708–3742).

Fig. R1 | A traditional example of olefin metathesis.

Question 3:

I am not well-familiar with the calculations, but the experiments show a clear difference between NHC–Cu and others metal complexes in terms of partial hydrogenation of acetylene. The calculations suggest, however, only the activity of NHC–Cu and selectivity for ethylene. What is needed, in my view, and what I think is buried in the calculations that have already been done, is a theoretical determination of the relative selectivity of NHC–Ag, NHC–Au, NHC–Pd as well as NHC–Cu.

Response:

We thank the Reviewer for above great comment. We do have a specific definition of selectivity in our DFT calculations. For the simulated reaction mechanism, there are two essential factors that will affect the selectivity. The first one (factor 1) is the competition between CH₃CH* and CH₂CH₂* formations. Another one (factor 2) is the competition between CH₂CH₂* desorption step and its deeper hydrogenation to CH₃CH₂*. For the factor 1, all catalysts had a more favorable formation of CH₂CH₂* than CH₃CH*, indicating that this factor did not affect their selectivity. In this case, the factor 2 dominates the selectivity. Clearly, the ideal catalyst with high selectivity needs to have a facile CH₂CH₂* desorption and simultaneous harsh over-hydrogenation to CH₃CH₂* species. As shown in Supplementary Figure 4, CH₂CH₂* desorption is exothermic by 0.07 eV on the NHC–Cu complex. However, this step for NHC–Ag, NHC–Au and NHC–Pd is endothermic by 0.2, 0.45 and 0.88 eV, respectively. In this respect, a more negative desorption energy indicates a higher selectivity for ethylene, i.e., NHC–Cu (–0.07) > NHC–Ag (0.2) > NHC–Au (0.45) > NHC–Pd (0.88).

The related discussions have been included in the revised manuscript (Page 7).

Specifically, I would like to see the initial state and the zero of energy to be the un-adsorbed gas phase reactant and the final state to be the desorbed product. This is important to evaluate the relative energetics between the electrocatalysts which will have different reactant adsorption energies.

Response:

We appreciate above valuable comment from the Reviewer. Indeed, the initial and final states of the reaction shown in Fig. 1 are already the gas phase reactant and desorbed product, denoted as CHCH(g), CH₂CH₂(g) and CH₃CH₃(g), respectively. In addition, we further provided the calculated total energy (E_{tot}), zero-point energy (E_{ZPE}) and entropic contributions (E_{S}) at 298.15 K of the gaseous reactants and products via RPBE functional (see Table R1). Note that our calculated total free energy for the acetylene hydrogenation to ethylene is -1.42 eV, which is very close to the experimentally detected value (-1.46 eV, data from the NIST standard reference database; <https://doi.org/10.18434/T4D303>), suggesting the high reliability of our DFT calculations.

Table R1. The calculated total energy (E_{tot}), zero-point energy (E_{ZPE}) and entropic (E_{S}) at 298.15 K of gaseous H₂, CHCH, CH₂CH₂ and CH₃CH₃ (the unit is eV).

Gas species	E_{tot}	E_{ZPE}	E_{S}
H ₂	-6.978	0.285	0.410
CHCH	-22.570	0.713	0.621
CH ₂ CH ₂	-31.669	1.350	0.678
CH ₃ CH ₃	-40.260	1.982	0.708

The related revisions have been included in Supplementary Table 2 in the revised supporting information (Page S3).

Question 4:

What is the reason to select alkaline electrolyte and is electrophilic/nucleophilic character of the substrates/intermediates/products are affected by the electrolyte (e.g., deprotonation at pH=14). The same question to the effect of electrolyte on electron-rich/poor character of the active sites as well as on σ -donation of the NHC ligand.

Response:

We appreciate the Reviewer for above constructive comments. The reason to select alkaline electrolyte is the optimal performance of NHC–Cu in 1 M KOH electrolyte. We have investigated electrocatalytic performance of NHC–Cu in different kinds of electrolytes in Fig. R2. NHC–Cu displayed a current density of 160 mA/cm² at –0.9 V vs. RHE in 1 M KOH, which was far larger than the values in 0.5 M KHCO₃ (39 mA/cm²), 0.1 M H₂SO₄ (20 mA/cm²) and 0.5 M H₂SO₄ (8 mA/cm²). Besides, NHC–Cu possessed the highest FE_{ethylene} of 98 % at –0.9 V vs. RHE in 1 M KOH with respect to 84.1 % in 0.5 M KHCO₃, 62.5 % in 0.1 M H₂SO₄ and 15.2 % in 0.5 M H₂SO₄. Clearly, along with decreased pH values of electrolytes, both of the activity and selectivity considerably decreased, which was attributed to competitive hydrogen evolution reaction. Declined pH values of electrolytes led to greatly elevated proton concentrations, which substantially benefited competitive hydrogen adsorption on Cu sites and suppressed acetylene adsorption. Accordingly, the alkaline electrolyte is selected for suppressing competitive hydrogen evolution.

The related revisions have been included in Supplementary Figure 19 in the revised supporting information (Page S20).

Fig. R2 | a, LSV curves of NHC–Cu in different electrolytes under the flow of pure acetylene at a scan rate of 1 mV s⁻¹. **b**, FE_{ethylene} of NHC–Cu in different electrolytes under flow of pure acetylene at –0.9 V.

Question 5:

What is the exact reason for a lower charge transfer resistance of NHC–Cu among the other cathodes?

Response:

The charge–transfer resistance (R_{ct}) obtained from the electrochemical impedance spectroscopy (EIS) is extensively used to estimate the electrode kinetics (*i.e.*, lower R_{ct} means faster reaction kinetics on electrode material surface). Combined with electrocatalytic performance and impedance spectroscopy of various electrocatalysts, the lower charge transfer resistance of NHC–Cu among other cathodes is thus attributed to its rapid reaction kinetics of electrocatalytic acetylene semihydrogenation (*Angew. Chem.*, 2018, 130, 1–5; *Angew. Chem.*, 2014, 126, 5252–5255; *ACS Catal.*, 2021, 11, 3257–3267; *ACS Catal.*, 2019, 9, 7398–7408).

Question 6:

Which materials has been used as GDE?

Response:

The carbon paper with a gas diffusion layer (Sigracet 29 BC, Fuel Cell Store) was used as gas diffuse electrode (GDE). Based on the comment, we have further defined the GDE in the revised manuscript (Page 14).

Question 7:

Did the authors study the leaching of the catalyst from GDE to the electrolyte solution while conducting stability testing?

Response:

As suggested by the Reviewer, we have studied the leaching of catalyst from GDE by using inductively coupled plasma mass spectrometry (ICP–MS). The ICP–MS

results revealed that the leaching of catalyst from GDE was negligible during the stability testing (Table R2).

Table R2. The ICP–MS results of NHC–Cu on GDE (25 cm² electrode area) before and after the long–term stability test.

Element	Conc. on GDE ($\mu\text{g}\cdot\text{cm}^{-2}$) before stability	Conc. on GDE ($\mu\text{g}\cdot\text{cm}^{-2}$) after stability
Cu	3.22	3.18
Cl	1.78	1.76

Question 8:

The main problem of the paper is that this paper is lack of discussion. As the authors mentioned in introduction, there are many different catalysts for gas–phase partial hydrogenation of acetylene with interesting performance known. Unfortunately, there is no information about electrocatalytic hydrogenation of acetylene. It is not clear if such electrochemical efforts are not exist or have been largely ignored by authors. The authors should show what are the new insights beyond the published results.

Response:

We thank the Reviewer for the insightful suggestion. According to the suggestion of the Reviewer, we have further included key discussions on electrocatalytic acetylene semihydrogenation in the revised introduction part as following: Nevertheless, the development of EAH falls far behind the conventional TAH as a result of poor solubility of acetylene in aqueous/organic solutions (1.06 g/kg H₂O) and strong competition of side reactions including hydrogen evolution reaction (HER), carbon–carbon coupling and overhydrogenation. Even using noble metal (e.g, Pd,¹⁶ Pt¹⁷ and Ag¹⁸) as electrocatalysts for pure acetylene, the EAH shows inferior ethylene selectivity (<70%), very low current densities (<3.5 mA·cm⁻²) and negligible space velocity.^{19, 20}

The related discussions have been included in the revised manuscript (Page 3).

Question 9:

The suppression of the competitive HER reaction should be discussed in more details.

Response:

We appreciate the Reviewer’s valuable suggestion. Accordingly, more detailed

descriptions on HER suppression were included in related experimental and theoretical parts. First, the density functional theory (DFT) simulations were conducted for evaluating the HER kinetics on the NHC–metal complexes. The related demonstration was added as “As shown in Supplementary Table S1, the Gibbs free energies of hydrogen (G_H) are -1.23 eV for NHC–Au, -1.13 eV for NHC–Cu, -0.82 eV for NHC–Ag, and 0.51 eV for NHC–Pd, which are far from zero. Especially, the large G_H of NHC–Cu (-1.13 eV) means too strong H–adsorption and sluggish HER kinetics on Cu site, which thus suppress the competitive HER reaction.” (Page 6 in the revised manuscript). Second, in experimental parts, the HER suppression was illustrated as “Notably, NHC–Cu retained a FE_{ethylene} value of ≥ 98 %, with a negligible FE_{C_4} and no FE_{ethane} over all applied potentials from -0.6 to -0.9 V (Fig. 2b and Supplementary Fig. 13). The total FE of the side reactions (HER, carbon–carbon coupling and overhydrogenation) on NHC–Cu was less than 2% over all applied potentials, indicating an excellent suppression effect towards side reactions. (Page 9)” and “During the 80–h EAH stability measurement at 30 mA/cm² (Fig. 3c), the FE_{ethylene} of NHC–Cu was consistently higher than 96%. Notably, the FE of hydrogen on NHC–Cu was less than 2% over the 80–h EAH test, indicating outstanding HER suppression. (Page 10)”

The related discussions on HER suppression have been included in the revised manuscript (Pages 6, 9 and 10).

Question 10:

The origin of hydrogen for electrochemical hydrogenation of acetylene should be discussed in more details as well.

Response:

As suggested by the Reviewer, we have further discussed the origin of hydrogen for electrocatalytic acetylene semihydrogenation as following: For the EAH, water rather than H₂ gas serves as a hydrogen source. In the cathode, water molecules dissociate and offer active hydrogen for in–situ hydrogenating acetylene ($C_2H_2 + 2H_2O + 2e^- \rightarrow C_2H_4 + 2OH^-$). (Page 3 in the revised manuscript).

Response to Reviewer 2:

Comments:

I have read the paper very much from the angle of trying to understand the underpinning chemistry rather than comment on the catalytic activity of the reported system relative to prior examples – other referees are better equipped to do that than me. I am left very confused by what the authors think is going on in their system by their continual reference to NHC–Cu, NHC–Ag, NHC–Au and NHC–Pd. In all cases, they start with NHC–M–chloride complexes; to my mind placing these in aqueous KOH would undoubtedly afford metal hydroxide complexes. Indeed, for an overview of this, see: D.J. Nelson, S.P. Nolan, *Coord. Chem. Rev.* 2017, 353, 278–294. This review and the primary literature shows that exactly these types of species have been produced for the specific compounds being studied. For Cu, see: G. C. Fortman, A. M. Z. Slawin, S. P. Nolan, *Organometallics* 2010, 29, 38966–3972. For Au, see: S. Gaillard, A. M. Z. Slawin, S. P. Nolan, *Chem. Commun.*, 2010, 46, 2742–2744. For a very close Pd analogue, see: J. D. Egbert, A. Chartoire, A. M. Z. Slawin, S. P. Nolan, *Organometallics* 2011, 30, 4494–4496.

Why is this important? Simply because copper is what the authors focus their attention on due to its high reactivity. A study of the (IPr)CuOH complex for efficient homogeneous catalytic alkyne semihydrogenation has been reported previously: N. O. Thiel, J. F. Teichert, *Org. Biomol. Chem.*, 2016, 14, 10660–10666. I cannot find any of the references above to show that the authors appreciate the fundamental chemistry that could help to explain their findings. They should repeat their studies using isolated examples of the (NHC)M(OH) precursors to show whether these display the same activity. Moreover, as apparent from the Nelson/Nolan review above, studies of stoichiometric alkyne deprotonation by e.g. (IPr)AuOH have been found to afford the corresponding (NHC)M(alkynyl) complexes. These too should be employed as catalytic precursors and their activity compared.

In short, the work should be rejected until the authors have taken on board the basic reactivity known in the literature to be associated with their catalyst precursors, appropriately cited the literature (the *Org. Biomol. Chem.* paper is particularly pertinent) and carried out tests of the activity of these (NHC)M(OH)/(NHC)M(alkynyl) species.

Until that is done, it is my view that the work is not suitable for publication anywhere as I believe it to be fundamentally flawed in its current format.

Response:

We greatly appreciate the Reviewer's valuable comments in view of underpinning chemistry. Accordingly, we have carefully investigated the literatures suggested by the Reviewer. The reported synthesis conditions of NHC–metal hydroxide complexes from NHC–metal chloride complexes are as follows: 1) for NHC–Cu hydroxide, 100 mg of NHC–Cu chloride complexes and **2 equiv of anhydrous CsOH** were first mixed in a scintillation vial of glovebox. Then, above solids were dissolved in **4.0 mL of dry, degassed THF** and further stirring for 8 h at room temperature. NHC–Cu hydroxide complexes was obtained **after the filter and concentration of resulting solution** (Organometallics, 2010, 29, 38966–3972). 2) For NHC–Au hydroxide, 100 mg of NHC–Au chloride complexes and **strong alkali (CsOH/NaOH/KOH, 0.322 mmol)** were first mixed in a scintillation vial and then dispersed into a mixture of **toluene and THF (1:1, 3.2 mL)**. Above mixture was stirred at **60 °C for 24 h** to synthesize the NHC–Au hydroxide complexes (Chem. Commun., 2010, 46, 2742–2744). 3) For NHC–Pd hydroxide, under **inert atmosphere**, (IPr)PdCl(cinnamyl)Cl (633.5 mg) and **CsOH (206 mg)** were mixed and then **30 ml of THF** was added. The reaction was stirring for 3 days to form NHC–Pd hydroxide complexes (Organometallics, 2011, 30, 4494–4496).

Obviously, as demonstrated in previous literatures, the synthesis of NHC–metal hydroxide complexes requires particularly harsh conditions: 1) a large dose of CsOH (strong base) is requisite; 2) a small amount of anhydrous and oxygen–free organic solvents (e.g., THF/Toluene) are necessary, which guarantees ultra–strong basic environment; 3) elevated reaction temperatures sometimes are needed, e.g., for NHC–Au hydroxide. Therefore, for the EAH in 1 M KOH aqueous solution at room temperature, such reaction conditions are substantially different from the harsh conditions for the synthesis of NHC–metal hydroxide complexes.

In order to experimentally investigate the structural stability during electrocatalytic acetylene semihydrogenation, we further conducted the ^1H NMR, XPS, Raman and inductively coupled plasma mass spectrometry (ICP-MS) characterizations of NHC-metal chloride complexes after 100-h dispersion in 1 M KOH aqueous solution and long-term electrocatalytic stability tests. As shown in Fig. R3a-b, ^1H NMR spectra of NHC-Cu chloride complexes after 100-h dispersion in 1 M KOH aqueous solution and long-term EAH tests were consistent with that for initial ones. In particular, no slightly shift of hydrogen in carbene backbone (CH-imide; e, e') was observed. After 100-h dispersion in 1 M KOH aqueous solution and long-term EAH tests, the XPS spectra of NHC-Cu also displayed no obvious variations (Fig. R3c). The atomic ratio of Cu and Cl in NHC-Cu retained well. As revealed in Fig. R4-6, similar to NHC-Cu, ^1H NMR and XPS spectra of NHC-Au, NHC-Ag and NHC-Pd demonstrated no detectable changes after their 100-h dispersion in 1 M KOH aqueous solution and long-term EAH tests. Specially, the ^1H NMR spectra of corresponding NHC-Pd hydroxide complexes principally possess upfield resonance at $-4.25/-4.29$ ppm, which corresponds to the hydrogen of bridging hydroxide moieties. However, these upfield resonance was not detected in ^1H NMR spectra of NHC-Pd after the 100-h dispersion in 1 M KOH aqueous solution and long-term EAH tests. No Raman stretching peak of OH was observed at $2800-3200\text{ cm}^{-1}$ after EAH test in 1 M KOH aqueous solution. (Fig. R7). Furthermore, the loading weights of Cu and Cl elements on the cathode were investigated by using the ICP-MS. The ICP-MS results revealed the contents of Cu and Cl elements remain well after the long-term stability test (Table R2). These results unambiguously confirm the structural stability of NHC-metal chloride complexes and exclude the formation of NHC-metal hydroxide and NHC-metal alkynyl complexes during the EAH.

In addition, we agree with the Reviewer. Some literatures focused on selective semihydrogenation of liquid-phase alkyne-contained molecules using organometallics as homogeneous catalysts. Such liquid-phase systems are substantially different from gas-phase acetylene semihydrogenation on heterogeneous catalysts (*Science*, 2018 362, 560-564; *Nat. Nanotechnol.*, 2018, 13, 856-861; *Nat. Mater.*, 2012, 11, 690-693;

Nat. Commun., 2019, 10, 4431), but they definitely provide important guidance on the design of high-performance heterogeneous catalysts. As suggested by the Reviewer, the reference (N. O. Thiel, J. F. Teichert, *Org. Biomol. Chem.*, 2016, 14, 10660–10666) has been cited as Ref. 30 in the revised manuscript.

Fig. R3 | a–b, ^1H NMR spectra and c, XPS spectra of pristine NHC–Cu (red line), NHC–Cu in 1 M KOH aqueous solution after 100 h (blue line) and NHC–Cu after long-term EAH stability test (black line).

Fig. R4 | **a–b**, ^1H NMR spectra and **c**, XPS spectra of pristine NHC–Au (red line), NHC–Au in 1 M KOH aqueous solution after 100 h (blue line) and NHC–Au after long–term EAH stability test (black line).

Fig. R5 | **a–b**, ¹H NMR spectra and **c**, XPS spectra of pristine NHC–Ag (red line), NHC–Ag in 1 M KOH aqueous solution after 100 h (blue line) and NHC–Ag after long–term EAH stability test (black line).

Fig. R6 | a–b, ^1H NMR spectra and c, XPS spectra of pristine NHC–Pd (red line), NHC–Pd in 1 M KOH aqueous solution after 100 h (blue line) and NHC–Pd after long-term EAH stability test (black line).

Fig. R7 | Raman spectra of a, NHC–Cu, b, NHC–Au, c, NHC–Ag and d, NHC–Pd before and after EAH test in 1 M KOH aqueous solution.

Response to Reviewer 3:**Comments:**

In this manuscript, the author introduced an N-heterocyclic carbene–Cu complex (NHC–Cu) as an efficient electrocatalyst for acetylene semihydrogenation reaction. Benefitting from the strong σ -donation from NHC, their catalyst exhibited outstanding activity, selectivity and durability for acetylene semihydrogenation reaction. This is an interesting attempt by employing electrons and protons to selectively convert C_2H_2 to C_2H_4 . The authors have performed comprehensive experiments to test this concept. However, major flaws exist. I would suggest its publication after a major revision to address the following comments.

Response:

We greatly appreciate the Reviewer's positive comments.

Question 1:

minor wording: you mentioned "High temperature" as a challenge for acetylene thermocatalytic semihydrogenation reaction (TAH). However, in literature, the reaction temperature for TAH is usually 80 – 100 °C. I don't think it's "high temperature". This needs to be reworded. Comparing with electrocatalysis at ambient Ts, TAH does need some thermal energy input; but definitely not "high T".

Response:

As illustrated by the Reviewer, the reaction temperature of several catalysts for TAH is 80–100 °C, but the optimal reaction temperature for most of thermocatalysts is 100–250 °C (Table 1 and Table R3). Particularly, for non-noble metal-based thermocatalysts, the reaction temperature is generally ≥ 200 °C for achieving efficient TAH (Table R3). Thus, by contrast with room temperature, we describe reaction temperature of thermocatalyst as high temperature based on a recent review (*Chem. Rev.*, 2020, 120, 683–733).

According to the suggestion of the Reviewer, "high temperature" has been revised to be "relatively high temperature" in the revised manuscript (Page 3).

Table R3. Summarized reaction temperatures for thermocatalytic acetylene semihydrogenation of catalysts

Catalyst	Temperature (°C)	References
Pd–Pt/SiO ₂	80	Science 362 , 560–564 (2018)
Pd–Zn/ZnO	80	ACS Catal. 6 , 1054–1061 (2016)
Pd/PPS	100	Sci. Adv. 6 , eabb7369 (2020)
PdZn–1.2@ZIF–8C	115	Adv. Mater. 30 , 1801878 (2018)
Pd@SOD	150	Angew. Chem. Int. Ed. 58 , 7668–7672 (2019)
CuPd single–atom	160	ACS Catal. 7 , 1491–1500 (2017).
Pd ₁ /ND@G	180	J. Am. Chem. Soc. 140 , 13142–13146 (2018)
Pd ₁ /CN	200	Nat. Nanotechnol. 13 , 856–861 (2018)
PdGa intermetallic	200	J. Am. Chem. Soc. 133 , 9112–9118 (2011)
Pd ₄ S/CNFs	250	J. Catal. 355 , 40–52 (2017)
NiGa	190	Angew. Chem. Int. Ed. 59 , 11647–11652 (2020)
Cu ₁ /ND@G	200	Nat. Commun. 10 , 4431 (2019)
Ni ₃ ZnCo _{0.7}	200	Nat. Commun. 11 , 3324 (2020)
Al ₁₃ Fe ₄	200	Nat. Mater. 11 , 690–693 (2012)
Ni@CeO ₂	200	J. Am. Chem. Soc. 140 , 12964–12973 (2018)
Co ₂ Mn _{0.5} Fe _{0.5} Ge	200	Sci. Adv. 4 , eaat6063 (2018)
Cu _{2.75} Ni _{0.25} Fe	250	J. Am. Chem. Soc. 132 , 4321–4327 (2010)
CeO ₂	250	Angew. Chem. Int. Ed. 51 , 8620–8623 (2012)
In ₂ O ₃	350	Angew. Chem., Int. Ed. 56 , 10755–10760 (2017).

Question 2:

Comments regarding the free energy diagram: it seems to me that according to the Bader Charge analysis, Pd–NHC should be more electrophilic, thus should adsorb C₂H₂ more easily. Why in the free energy diagram, the first step of C₂H₂ adsorption, it's not the most down–hill on Pd–NHC?

Response:

We appreciate the great comment from the Reviewer. In fact, C₂H₂ adsorption is the most down–hill on Pd–NHC as shown in Fig. 1, which exactly agrees with the Reviewer's insightful knowledge about the electrophilic feature of Pd. Note that Bader charge data in Fig. 1b does not reflect the intrinsic electrophilic properties of the metals, it only shows the amount of charges that transfer from the NHC to the metals.

What is the RDS on M–NHC? The first adsorption step is not favored on Cu–NHC while the last desorption step of C₂H₄ seems the most facile on Cu–NHC. This calculation result combined with the experimental observation of the best performance of Cu–NHC may suggest the desorption might be the RDS. However, it needs some evidence and clarification. It seems kinetic experiments have been performed but not be analyzed in–depth to provide insight into the RDS. I would suggest analyzing the kinetics more carefully and obtain reaction rates/RDS which can be combined with DFT calculations.

Response:

We greatly appreciated the constructive suggestion of the Reviewer. As pointed out by the reviewer, the DFT calculations revealed that ethylene desorption from NHC–Cu was thermodynamically favorable, which played a key role in obtaining high ethylene selectivity. As suggested, we further conducted in–situ electrochemical Raman investigations on NHC–Au, NHC–Ag and NHC–Pd (Fig. R10). When the potential of NHC–Au reached –0.2 V, two characteristic peaks appeared at 1122 and 1495 cm^{–1} (Fig. R10a), which were assigned to symmetric CH₂ scissors and C=C stretch modes of adsorbed ethylene, respectively. Similarly, the peaks of adsorbed ethylene on NHC–Ag appeared gradually at 1122 and 1507 cm^{–1} when the potential was increased from 0 V to –0.8 V (Fig. R10b). For NHC–Pd, adsorbed ethylene peaks of 1120 and 1507 cm^{–1} was observed once the potential reached –0.2 V (Fig. R10c). Obviously, all C=C stretch modes of NHC–Au, NHC–Ag and NHC–Pd displayed a negative shift versus 1547 cm^{–1} for NHC–Cu. Moreover, the characteristic peaks of ethylene still exist for NHC–Au, NHC–Ag and NHC–Pd when the EAH was terminated. These results unambiguously proved the weak absorption of ethylene on NHC–Cu relative to other catalysts, which was consistent with theoretical results.

We used the electrochemical impedance spectroscopy (EIS) to estimate the reaction kinetics (Supplementary Figure 14). In comparison with other NHC–metals, lower charge transfer resistance (~45 Ω) of NHC–Cu indicated rapid reaction kinetic. For providing deep insights into kinetic reaction rate, we calculated the Tafel slopes of NHC–metals (Fig. R8). NHC–Cu showed a Tafel slope of ~199 mV/dec, which was

lower than 203 mV/dec for NHC–Ag, 212 mV/dec for NHC–Pd and 238 mV/dec for NHC–Au, indicating an accelerated reaction rate of NHC–Cu for electrocatalytic acetylene semihydrogenation.

In principle, the EAH involves four steps: 1) acetylene adsorption; 2) first hydrogenation of C_2H_2 to C_2H_3 ; 3) second hydrogenation of C_2H_3 to C_2H_4 ; 4) ethylene desorption. In the past, the rate-determining step of acetylene semihydrogenation is elusive and challenging. In this work, for NHC–Cu and NHC–Pd, the reaction kinetics of acetylene adsorption (Fig. 1c), first hydrogenation and second hydrogenation steps were very similar. However, ethylene desorption was only favorable on NHC–Cu, which accorded well with experimental results. Accordingly, for the EAH, the ethylene desorption process might be the rate-determining step.

The related contents have been included in the revised supplementary information (Supplementary Figure 15–16 in Page S15–17).

Fig. R8 | The Tafel slopes of NHC–Cu, NHC–Ag, NHC–Pd and NHC–Au.

The reaction profile seems to be solely based on thermodynamic calculations without considering electrode polarization. I would suggest adding a free energy diagram with potential consideration.

Response:

We greatly appreciated the constructive suggestion of the Reviewer. We have added a free energy diagram at a potential of -0.9 V vs. RHE (Fig. R9). Note that NHC–Cu is still the best for acetylene semihydrogenation among the four electrocatalysts at -0.9 V

vs. RHE.

The related contents have been included in the revised supplementary information (Supplementary Figure 4e, Page S5).

Fig. R9 | Free energy diagram for acetylene semihydrogenation on different electrocatalysts at a potential of -0.9 V vs. RHE.

Question 3:

Comments on the electrocatalysis experimentation:

The Raman spectroscopy of Cu–NHC can be compared with other M–NHC just to confirm whether the assumed adsorption/desorption and the difference among M–NHCs echo with the DFT calculations. A side note: Why does the peak at 1954 (indication of C₂H₂ adsorption) disappear after -0.4 V?

Response:

We thank the reviewer for valuable suggestion. We further conducted in-situ electrochemical Raman investigations on NHC–Au, NHC–Ag and NHC–Pd (Fig. R10). The corresponding discussions have been included as following: When the potential of NHC–Au reached -0.2 V, two characteristic peaks appeared at 1122 and 1495 cm⁻¹ (Fig. R10a), which were assigned to symmetric CH₂ scissors and C=C stretch modes of adsorbed ethylene, respectively. Similarly, the peaks of adsorbed ethylene on NHC–Ag appeared gradually at 1122 and 1507 cm⁻¹ when the potential was increased from 0 V to -0.8 V (Fig. R10b). For NHC–Pd, adsorbed ethylene peaks of 1120 and 1507 cm⁻¹ was observed once the potential reached -0.2 V (Fig. R10c). Obviously, all C=C stretch modes of NHC–Au, NHC–Ag and NHC–Pd displayed a negative shift versus 1547 cm⁻¹ for NHC–Cu. Moreover, the characteristic peaks of ethylene still exist for NHC–Au,

NHC–Ag and NHC–Pd when the EAH was terminated. These results unambiguously proved the weak absorption of ethylene on NHC–Cu relative to other catalysts, which was consistent with theoretical results.

The disappearance of acetylene peak at 1954 cm^{-1} after -0.4 V was attributed to the increased electrocatalytic acetylene semihydrogenation activity at high potentials, which rapidly consumed the adsorbed acetylene molecules on NHC–Cu.

The related revisions have been included in Supplementary Figure 16 in the revised supplementary information (Page S16–17).

Fig. R10 | In situ electrochemical Raman spectra of (a) NHC–Au, (b) NHC–Ag and (c) NHC–Pd in a 1 M KOH aqueous solution. For clarity, the spectral regions of 1020–1190 cm^{-1} and 1400–1610 cm^{-1} were expanded.

How are those current densities normalized? Is it based on geometric surface area or ECSA?

Response:

The current density was normalized with the geometric area of the electrode. For EAH tests, the loading weight of the NHC–metal complexes was only 25 $\mu\text{g}\cdot\text{cm}^{-2}$. So, it is too difficult to normalize current densities with specific surface area or ECSA of NHC–metal complexes.

The related contents have been included in the Electrochemical measurements in the revised manuscript (Page 16).

For the “1 % acetylene impurities (1×10^4 ppm) in crude ethylene” test, I noticed the flow rate of feeding gas is 10 sccm. However, in literature, THA catalysts typically work at a flow rate of 50–100 sccm. Have you tried a higher flow rate of feed gas here? The selectivity probably won’t change much but I am wondering whether the HNC–Cu system can still keep that high conversion.

Response:

As demonstrated by the Reviewer, thermocatalytic acetylene hydrogenation (TAH) generally works at a flow rate of 50–100 $\text{mL}\cdot\text{min}^{-1}$. However, the TAH was always conducted in the feed gas containing abundant He/N₂/Ar as the balance gas. Meanwhile, the flow rate is related with volume of reactors and loading weight of catalysts, which together correspond to space velocity (a key parameter).

Based on the Reviewer’s suggestion, for crude ethylene containing 1 % acetylene, we have tried higher flow rates. As shown in Fig. R11, the NHC–Cu can achieve acetylene conversion of 99.4 %, 97.5 %, 96.7% and 94.1 % respectively at increased flow rates of 20, 30, 40 and 50 $\text{mL}\cdot\text{min}^{-1}$, but the specific selectivities were always >99%.

The related results have been included in the revised supplementary information

(Supplementary Figure 28 in Page S29).

Fig. R11 / The conversion and specific selectivity of acetylene versus cathodic current in 1 M KOH aqueous solution using NHC–Cu at different flow rate of crude ethylene containing 1 % acetylene: (a) $20 \text{ mL}\cdot\text{min}^{-1}$, (b) $30 \text{ mL}\cdot\text{min}^{-1}$, (c) $40 \text{ mL}\cdot\text{min}^{-1}$ and (d) $50 \text{ mL}\cdot\text{min}^{-1}$.

What about the solubility issue? Will the solubility of C_2H_2 in water become a concern for the electronic semi-hydrogenation of C_2H_4 ? This is an issue where THA catalysts do not suffer from.

Response:

We thank the Reviewer’s constructive comment. The poor solubility of acetylene in electrolytes (1.06 g/kg H_2O) will seriously limit the electrocatalytic acetylene hydrogenation performance using H-cell as reactors, where acetylene is bubbled into the electrolytes. However, in this work, we employed customized flow-cell with abundant three-phase interfaces (acetylene gas–solid catalysts–liquid electrolyte) as the EAH reactor, which intrinsically overcame the limitation of acetylene solubility/diffusion in electrolyte.

REVIEWER COMMENTS

Reviewer #1 (Remarks to the Author):

Thank you for addressing my comments. The paper has been improved and now can be published.

Reviewer #2 (Remarks to the Author):

The authors are thanked for their comprehensive replies to the reviewer comments. They have made reference to the literature regarding the formation of (NHC)Cu(OH) complexes and done some measurements that they feel prove that such a species cannot be involved; firstly, this is down to what they regard as them employing milder reaction conditions than those used in the used in the literature to prepare (IPr)Cu(OH). Secondly, they compare NMR spectra - the literature shows a 0.05-0.06 ppm difference in the chemical shifts of the resonances between the Cu-Cl and Cu-OH species (Chem. Commun., 2013, 49, 10483-10485 vs. Organometallics 2010, 29, 3966–3972). Such a shift would be very hard to differentiate, so I'm not convinced this part of their rebuttal holds. I know very little about XPS to say what %error there needs to be in the C:Cl ratio in order for it to be said to have changed.

I would be far more reassured if they were to use (IPr)Cu(OH) as a pre-catalyst for a direct comparison to their Cu-Cl system. This would be very informative, especially if it proved to be the same in terms of activity, or even either less or more active. The authors may consider my views pedantic with regard to what they view as the bigger picture and story they are trying to report, but I think it is down to them to prove their case to the referees convincingly, especially when trying to get the work published in such a high impact journal. However, in a more general sense, irrespective of where they are aiming to publish, it is simply fundamental in helping to rationalize the catalysis that there is some understanding of mechanism, if only to allow other researchers to build upon the findings. I, and I'm sure many other researchers trying to bring about the activation of small organic molecules like alkynes, do not see how (IPr)CuCl can react with $\text{HC}\equiv\text{CH}$, but I can see how (IPr)Cu(OH) could, based on simple literature precedent.

I stand by my previous comment that the work is 'fundamentally flawed' without tests on the activity of (IPr)Cu(OH).

Reviewer #3 (Remarks to the Author):

The authors have addressed my questions well. I would suggest its publication in the current form.

Response to Reviewer 1:**Comments:**

Thank you for addressing my comments. The paper has been improved and now can be published.

Response:

We greatly appreciate the Reviewer's positive comments.

Response to Reviewer 2:**Comments:**

The authors are thanked for their comprehensive replies to the reviewer comments. They have made reference to the literature regarding the formation of (NHC)Cu(OH) complexes and done some measurements that they feel prove that such a species cannot be involved; firstly, this is down to what they regard as them employing milder reaction conditions than those used in the used in the literature to prepare (IPr)Cu(OH). Secondly, they compare NMR spectra – the literature shows a 0.05 – 0.06 ppm difference in the chemical shifts of the resonances between the Cu – Cl and Cu – OH species (Chem. Commun.,2013, 49, 10483 – 10485 vs. Organometallics 2010, 29, 3966 – 3972). Such a shift would be very hard to differentiate, so I'm not convinced this part of their rebuttal holds. I know very little about XPS to say what %error there needs to be in the C:Cl ratio in order for it to be said to have changed.

I would be far more reassured if they were to use (IPr)Cu(OH) as a pre – catalyst for a direct comparison to their Cu – Cl system. This would be very informative, especially if it proved to be the same in terms of activity, or even either less or more active. The authors may consider my views pedantic with regard to what they view as the bigger picture and story they are trying to report, but I think it is down to them to prove their case to the referees convincingly, especially when trying to get the work published in such a high impact journal. However, in a more general sense, irrespective of where they are aiming to publish, it is simply fundamental in helping to rationalize the catalysis that there is some understanding of mechanism, if only to allow other researchers to build upon the findings. I, and I'm sure many other researchers trying to bring about the activation of small organic molecules like alkynes, do not see how

(IPr)CuCl can react with HC≡CH, but I can see how (IPr)Cu(OH) could, based on simple literature precedent.

I stand by my previous comment that the work is 'fundamentally flawed' without tests on the activity of (IPr)Cu(OH).

Response:

We greatly thank the Reviewer for the insightful suggestion again. Accordingly, we have further synthesized the compound (IPr)Cu(OH) (named as NHC–Cu(OH)) based on the reference suggested by the Reviewer (Organometallics, 2010, 29, 3966–3972). Specifically, 100 mg 1,3-bis(2,6-diisopropylphenyl)imidazol-2-ylidene]copper chloride (denoted as NHC–Cu) was added into a dried round bottle (25 mL) and transferred to the glovebox. After adding anhydrous cesium hydroxide (CsOH, 61 mg) and dry tetrahydrofuran (THF, 4 mL) into above bottle, the mixture was reacted at room temperature for 8 h under N₂ atmosphere. The resulting solution was filtered through a plug of Celite and concentrated in vacuo until a white precipitate formed (ca. 1 mL remaining). The 1,3-bis(2,6-diisopropylphenyl)imidazol-2-ylidene (hydroxy) copper (denoted as NHC–Cu(OH)) was finally collected after the precipitation in hexane and dried under vacuum. As displayed in Fig. R1–R4, ¹H–NMR spectrum was performed to characterize the chemical structure of NHC–Cu and NHC–Cu(OH).

¹H–NMR of [NHC–Cu(OH)] (500 MHz, CD₂Cl₂): δ 7.52 (t, *J* = 8.0 Hz, 2H), 7.33 (d, *J* = 8.0 Hz, 4H), 7.13 (s, 2H), 2.57 (p, *J* = 7.0 Hz, 4H), 1.27 (d, *J* = 7.0 Hz, 12H), 1.21 (d, *J* = 7.0 Hz, 12H).

¹H–NMR of [NHC–Cu] (500 MHz, CD₂Cl₂): δ 7.54 (t, *J* = 8.0 Hz, 2H), 7.35 (d, *J* = 8.0 Hz, 4H), 7.19 (s, 2H), 2.57 (p, *J* = 7.0 Hz, 4H), 1.29 (d, *J* = 7.0 Hz, 12H), 1.23 (d, *J* = 7.0 Hz, 12H).

Notably, the ¹H–NMR spectrum of NHC–Cu possessed protons of imid–CH at 7.19 ppm in the carbene backbone. Followed by the formation of NHC–Cu(OH), the peak of imid–CH at 7.19 ppm shifted to a lower field (7.13 ppm, Fig. R4), which was consistent with the difference of chemical shifts (0.05–0.06 ppm) between Cu – Cl and Cu – OH species (Chem. Commun., 2013, 49, 10483 – 10485; Organometallics 2010, 29, 3966 – 3972).

Fig. R1 | ¹H-NMR spectrum of NHC-Cu(OH) in deuterated dichloromethane.

Fig. R2 | ¹H-NMR spectrum of NHC-Cu in deuterated dichloromethane.

Fig. R3 | The comparison of ^1H -NMR spectrum of NHC-Cu and NHC-Cu(OH) in deuterated dichloromethane.

Fig. R4 | The aromatic region of ^1H -NMR spectrum of NHC-Cu and NHC-Cu(OH) in deuterated dichloromethane.

Afterwards, we have systematically evaluated electrocatalytic acetylene semihydrogenation performance of NHC–Cu(OH) for comparison with that of NHC–Cu. As shown in Fig. R6a, the current density of NHC–Cu(OH) shows a dramatically negative shift from 162 mA/cm² for NHC–Cu to 98 mA/cm² at –0.9 V vs. RHE, demonstrating the outstanding electrocatalytic activity of NHC–Cu. Meanwhile, NHC–Cu retained the FE_{ethylene} of ≥98 % over potentials from –0.6 to –0.9 V. However, NHC–Cu(OH) produced increased H₂ along with decreased potentials (Fig. R5). Especially at a large potential of –0.9 V (Fig. R6b), the FE_{ethylene} of NHC–Cu(OH) obviously decreased to 66% relative to 98% for NHC–Cu. Furthermore, NHC–Cu achieved a partial ethylene current density of up to 159 mA/cm² at –0.9 V (Fig. 6c), which was substantially larger than 65 mA/cm² for NHC–Cu(OH). These results not only confirmed the excellent performance of electrocatalytic acetylene semihydrogenation on NHC–Cu in comparison with NHC–Cu(OH), but also reveal the important role of other coordinated ligands for modulating the electrocatalytic performance of NHC–metal. We thank the Reviewer again for such meaningful guidance for our future work.

Fig. R5 | FEs of gaseous products in 1 M KOH aqueous solution under the flow of pure acetylene by using **a**, NHC–Cu and **b**, NHC–Cu(OH) as electrocatalysts.

Fig. R6 | a, Polarization curves of NHC–Cu complexes (NHC–Cu and NHC–Cu(OH)) in a 1 M KOH aqueous solution under pure acetylene flow. **b,** Ethylene FEs of NHC–Cu complexes in a 1 M KOH aqueous solution at different potentials under pure acetylene flow. **c,** Partial current density of ethylene for NHC–Cu complexes in a 1 M KOH aqueous solution at different potentials under pure acetylene flow.

The related revisions have been included in Page 11, 15, 26 in the revised manuscript and Supplementary Figure 25–30 in the revised Supporting Information (Page S26–S29).

Response to Reviewer 3:

Comments:

The authors have addressed my questions well. I would suggest its publication in the current form.

Response:

We greatly appreciate the Reviewer's positive comments.

REVIEWERS' COMMENTS

Reviewer #2 (Remarks to the Author):

The authors have taken on board the suggestion of investigating the possible role of (IPr)Cu(OH) in their findings. The results presented in the new manuscript and ESI show that it has reduced activity relative to what they see with the Cu-Cl precursor. It has to be said that this leaves me baffled as to how their chemistry therefore actually works with regard to interaction with alkyne and subsequent semi-reduction, but they have done all that has been asked of them as far as pursuing these test experiments. I hope future studies will shed more light on the mechanism of operation, but for now, I am prepared to recommend publication of their manuscript in Nature Communications.

One minor correction is needed on page 15 on lines 251-254: it should be.....diisopropylphenyl)imidazol-2-ylidene not diisopropylphenyl)imidazole-2-ylidene.

Response to Reviewer 2:**Comments:**

The authors have taken on board the suggestion of investigating the possible role of (IPr)Cu(OH) in their findings. The results presented in the new manuscript and ESI show that it has reduced activity relative to what they see with the Cu-Cl precursor. It has to be said that this leaves me baffled as to how their chemistry therefore actually works with regard to interaction with alkyne and subsequent semi-reduction, but they have done all that has been asked of them as far as pursuing these test experiments. I hope future studies will shed more light on the mechanism of operation, but for now, I am prepared to recommend publication of their manuscript in Nature Communications.

Response:

We greatly appreciate the Reviewer's positive comments.

One minor correction is needed on page 15 on lines 251-254: it should be.....diisopropylphenyl)imidazol-2-ylidene not diisopropylphenyl)imidazole-2-ylidene.

Response:

We are sorry about the mislabeling of “.....diisopropylphenyl)imidazole-2-ylidene” on page 15. We have updated the manuscript as suggested by the Reviewer.